# A robust platform streamlining aromatic noncanonical amino acid biosynthesis and genetic code expansion in *Escherichia coli*

Jingxuan Zhang[1,2,8], Keying Yu[1,3,8], Yali Xu[4,8], Wushuang Zhao[1,5], Yulian Li[1,5], Ying Wang[2], Florian P. Seebeck [6], Xiao-Hua Chen [4,7] ✉ & Cangsong Liao [1,3,5] ✉

Genetic code expansion (GCE) has significantly enhanced the diversity of proteins in the biological world, leading to a wide range of applications. Despite the advances in GCE, the cost of noncanonical amino acids (ncAAs) remains one of the major obstacles for large-scale production. In situ biosynthesis of ncAAs from commercial precursors offers a promising solution to this challenge, yet only a few biosynthetic pathways have been reported. Here, we present a platform that couples the biosynthesis of aromatic ncAAs with genetic code expansion in *E. coli*, enabling the production of proteins and peptides containing ncAAs. Forty ncAAs are synthesized from aryl aldehydes by the biosynthetic pathway, while nineteen ncAAs are incorporated into superfolder GFP using three orthogonal translation systems. The platform's versatility is demonstrated by the production of macrocyclic peptides and antibody fragments. We envision that the platform will facilitate the production of peptides, enzymes, and antibody fragments containing ncAAs.

Proteins produced by living organisms in nature are primarily composed of 20 canonical amino acids. While the chemical space and functionality of natural proteins are constrained by the diversity of these amino acids, their properties can be expanded and modified through posttranslational modifications. Site-specific incorporation of noncanonical amino acids (ncAAs) via genetic code expansion (GCE) enables the production of proteins with enhanced or novel properties[1–4], thus facilitating the creation of a broad range of proteins with potential applications in analytical chemistry[5], catalysis[6,7], biological science[8], materials science[9], and medicines[10,11].

To date, over 300 ncAAs with diverse functional groups have been successfully utilized in GCE, and this number continues to grow[12,13]. Beyond classic L-α-amino acids, engineered aaRS-tRNA pairs have been developed to incorporate other noncanonical building blocks[14], such

as α-hydroxy acids[15], β-amino acids and α,α-disubstituted amino acids[16], for site-specific incorporation in target proteins. Furthermore, substantial progress has been made in enhancing the efficiency of ncAA incorporation through various improvements in GCE technology. For instance, the incorporation efficiency of ncAAs at the amber codon has been enhanced by the deletion of release factor 1[17,18]. Additionally, the efficiency and robustness of ncAA incorporation have been further optimized through the engineering of aaRSs and ribosomal machinery using diverse evolutionary approaches[19]. Another strategy to improve the efficiency of ncAAs incorporation is the recoding of rare codons in both mammalian cells[20] and synthetic *Escherichia coli* strains[21,22].

While significant progress has been made in many aspects of GCE technology, particularly in the engineering of aminoacyl-tRNA

[1]State Key Laboratory of Chemical Biology, Shanghai Institute of Material Medica, Chinese Academy of Sciences, Shanghai, China. [2]State Key Laboratory of Synthetic Biology, Tianjin University, Tianjin, China. [3]School of Chinese Materia Medica, Nanjing University of Chinese Medicine, Nanjing, China. [4]State Key Laboratory of Drug Research, Shanghai Institute of Materia Medica, Chinese Academy of Sciences, Shanghai, China. [5]University of Chinese Academy of Sciences, Beijing, China. [6]Department for Chemistry, University of Basel, Basel, Switzerland. [7]School of Pharmaceutical Science and Technology, Hangzhou Institute for Advanced Study, University of Chinese Academy of Sciences, Hangzhou, China. [8]These authors contributed equally: Jingxuan Zhang, Keying Yu, Yali Xu. ✉e-mail: xhchen@simm.ac.cn; csliao@simm.ac.cn

synthetases (aaRSs) to improve the incorporation of ncAAs, the supply of ncAAs remains a major challenge. This issue is often referred to as "the Achilles' heel" of GCE technology, hindering the development of large-scale commercial applications[23]. In conventional GCE-based experiments (Fig. 1a), ncAAs must be supplied exogenously as a medium supplement at concentrations of 1–10 mM for orthogonal translation systems (OTSs) during protein production. Many high-value ncAAs are either not commercially available or too expensive for large-scale protein production, as producing enantiomerically pure ncAAs in sufficient quantities remains a challenge in organic synthesis. Additionally, some ncAAs exhibit low membrane permeability, preventing efficient uptake into cells and resulting in reduced protein yields[24]. Therefore, coupling the biosynthesis of the required ncAAs with GCE within the same host cell offers a practical solution to this problem[25]. In 2003, the Schultz group demonstrated the first proof-of-concept for the biosynthesis of 4-aminophenylalanine (**b1**) from simple carbon sources and its site-specific incorporation into proteins[26]. Since then, various examples of de novo biosynthesis of ncAAs for GCE through metabolic engineering have been reported, such as O-phospho-L-threonine[24], norleucine[27], 5-hydroxytryptophan (**b4**)[28], O-methyltyrosine (**b3**)[29], sulfotyrosine[30], and 4-nitrophenylalanine (**b2**)[31] (Representative structures in Fig. 1b). However, the de novo biosynthesis of ncAAs faces several challenges, such as the lack of known pathways for most ncAAs, low efficiency, and limited diversity of known pathways.

In contrast, semisynthesis of ncAAs offers more efficient and versatile solutions for providing ncAAs with various structures for incorporation into proteins (Fig. 1c). For example, the pyrrolysine biosynthetic pathway has been reprogrammed for the synthesis of pyrrolinecarboxy-lysine (**c3**), ethynylpyrrolysine and D-cysteinyl-Nᵉ-L-lysine[32]. L-dihydroxyphenylalanine (**c2**) has been synthesized from catechol, pyruvate, and ammonia by tyrosine phenol-lyase[33]. Additionally, S-allyl-L-cysteine[34] and a series of S-aryl-L-cysteine (**c1**)[35] have been synthesized from allyl mercaptan and aromatic thiols by hijacking the cysteine biosynthetic pathway and successfully incorporated into proteins in E. coli cells. Despite these pioneering studies, the diversity of semisynthetically produced ncAAs is still significantly limited compared to that of ncAAs incorporated via GCE. Only a few biosynthetic pathways have been constructed for the widely used aromatic ncAAs and coupled with GCE. One related study by Lee and colleagues demonstrated the biosynthesis of phenylalanine derivatives from corresponding α-keto acid precursors[36,37]. They introduced an aminotransferase from Thermus thermophilus into E. coli and genetically encoded the system using the MmPylRS/tRNA$^{Pyl}_{CUA}$ pair. However, α-keto acids are generally more expensive than their corresponding amino acids, making them less ideal for large-scale GCE applications.

In this work, we aimed to develop a generic in vivo biosynthetic pathway and construct a semiautonomous E. coli strain for the efficient synthesis of aromatic ncAAs, which can be directly utilized by orthogonal translation systems (OTSs) within the same cell. We envisioned that an ideal pathway for this purpose should meet the following criteria. First, the pathway should involve a minimal number of enzymatic steps (preferably fewer than five), with enzymes that have high reaction rates and are capable of efficiently producing ncAAs at titers suitable for downstream incorporation by OTSs. Second, the starting materials for the pathway should be abundant, commercially available at low cost, and possess diverse functional groups. Third, the enzymes

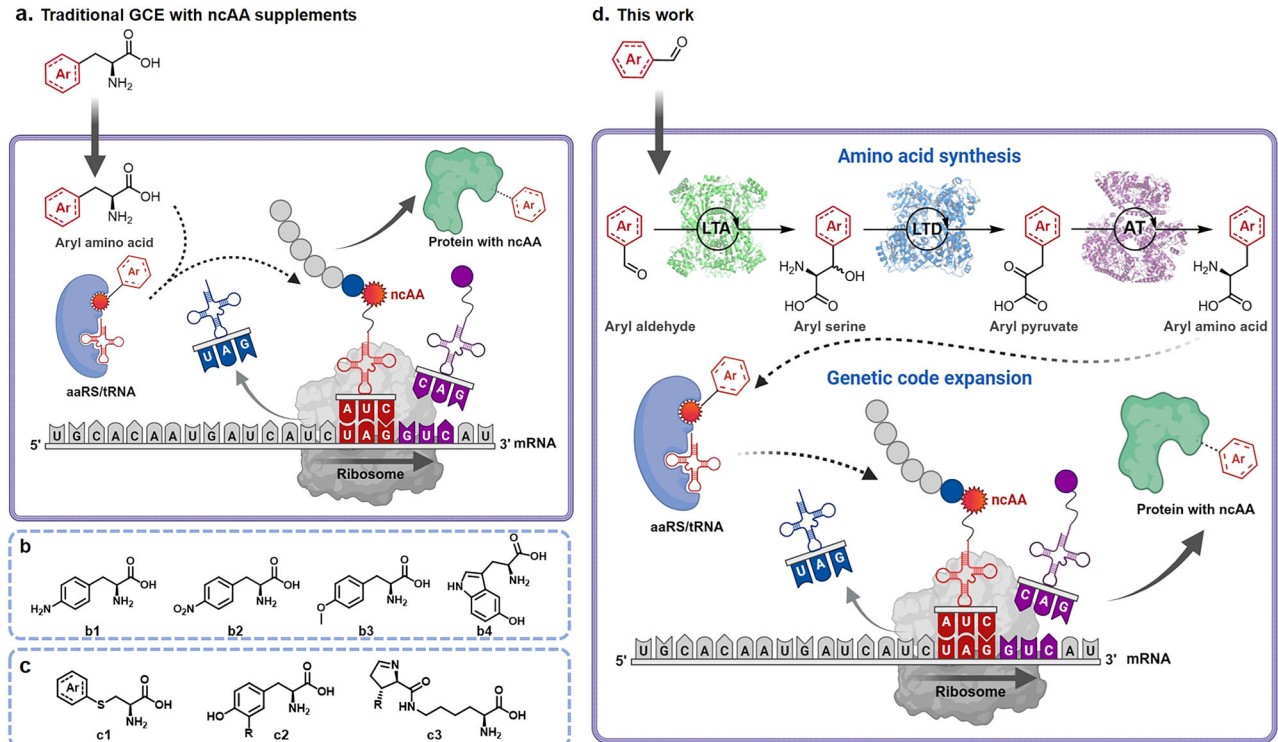

**Fig. 1 | E. coli strain for synthesis of noncanonical amino acids and application to genetic code expansion. a** Traditional GCE with ncAAs supplements. The orthogonal translation system relies on the extrinsic addition of ncAAs to the medium for protein production. Created in BioRender (https://BioRender.com/cf6rx1z). **b** Representative ncAAs by de novo biosynthesis for GCE through metabolic engineering. **c** Representative ncAAs by semisynthesis from precursors for GCE. **d** The engineered E. coli strain developed in this study utilizes a multi-enzyme cascade pathway to convert a range of aromatic aldehydes into L-aromatic ncAAs.

This pathway involves the overexpression of threonine aldolase (LTA), threonine deaminase (LTD), and the native aromatic amino acid aminotransferase (AT) in E. coli. The synthesized ncAAs are then incorporated into the target protein through genetic code expansion (GCE) in situ, facilitated by the expression of exogenous aminoacyl-tRNA synthetase (aaRS) and tRNA. This approach establishes an engineered strain for both ncAA biosynthesis and the production of ncAA-containing proteins. Created in BioRender (https://BioRender.com/9aei6t2).

in the pathway should be promiscuous, able to accept a wide range of substrates to produce ncAAs with varied functionalities, thereby facilitating their incorporation by GCE and enabling further applications.

Based on these principles, we report here a general approach that couples the biosynthesis of aromatic ncAAs with GCE for the efficient production of site-specifically modified recombinant proteins in *E. coli* (Fig. 1d). Through pathway design, enzyme screening, and host engineering, we develop a bacterial strain that can produce 40 different aromatic amino acids in vivo starting with the corresponding aromatic aldehydes. Nineteen of these ncAAs are successfully incorporated into target proteins in situ by GCE using three classic OTSs in *E. coli*. This method, along with the semiautonomous *E. coli* strain, provides a generic, efficient, and cost-effective platform for the large-scale recombinant production of proteins of interest.

## Results

### Design of the biosynthetic pathway

We designed two biosynthetic pathways for the synthesis of ncAAs, one pathway starting from aryl propionic acids (path 1), one from aryl aldehydes (path 2) (Supplementary Fig. 1). Both pathways are based on retro-biosynthesis principles and consist of three enzymatic steps, with a common transamination reaction as the final step. The first pathway features a key dioxygenase that catalyzes the C$\alpha$-H oxyfunctionalization of aryl propionic acids, producing the corresponding aryl lactate. These $\alpha$-hydroxy acids are then oxidized by a dehydrogenase to aryl pyruvate, which serves as precursors for the ncAAs. As part of an earlier study, we have repurposed and engineered $\alpha$-ketoglutarate-dependent dioxygenases for the synthesis of L-aryl lactate[38,39]. These dioxygenases exhibited high activity and broad substrate scope in vitro. However, despite their promising performance in vitro, the dioxygenase variants were not efficient enough for in vivo applications and most of the substrates were not converted in the culture. The reaction with (2-(4-iodophenyl)propanoic acid was shown as a representative in Supplementary Fig. 2a. In addition, the supplementation of the reaction mixtures was found to completely inhibit the growth of *E. coli* in our preliminary experiment (Supplementary Fig. 2b).

To avoid these two problems, we considered pathway 2 (Fig. 1d), in which benzaldehydes are converted to ncAAs in vivo. The first step in this pathway involves an aldol reaction between glycine and aryl aldehyde, catalyzed by L-threonine aldolase (LTA, EC 4.1.2.5), resulting in the production of aryl serines. The intermediates are then converted by L-threonine deaminase (LTD, EC 4.3.1.19) into aryl pyruvates, which serve as a substrate for the final transamination step, catalyzed by an aminotransferase, to yield the ncAAs. In previous enzymatic cascade experiments, we demonstrated that LTA and LTD efficiently convert a variety of benzaldehyde derivatives into their corresponding $\alpha$-keto acids[40]. Additionally, we anticipated that the promiscuous aromatic amino acid aminotransferase (TyrB) in *E. coli* would effectively catalyze the third step, given TyrB's high catalytic efficiency ($k_{cat}/K_m$ up to 1250000 M$^{-1}$ s$^{-1}$)[41] and its broad scope[42].

### Initial demonstration with *p*-iodophenylalanine

We began our experiments by assessing the activity of the pathway in vitro. To this end, we recombinantly expressed and purified phenylserine aldolase from *Pseudomonas putida* (PpLTA), threonine deaminase from *Rahnella pickettii* (RpTD), and TyrB, enabling us to perform the cascade reactions. Using 1 mM concentrations of *para*-iodobenzaldehyde and *para*-methylbenzaldehyde, the enzyme cascade efficiently converted the substrates into their corresponding amino acids within 0.5 and 2 h, respectively (Supplementary Fig. 3).

We next evaluated the activity of the pathway in a lyophilized *E. coli* whole-cell catalyst format. A new strain, *E. coli* BL21 (PpLTA-RpTD), was constructed from *E. coli* BL21 (DE3) by transforming a pACYCDuet-1 vector expressing the genes encoding PpLTA and RpTD.

As shown in Supplementary Fig. 4, 5 mg/mL lyophilized *E. coli* BL21 (PpLTA-RpTD) produced 0.96 mM *p*-iodophenylalanine (pIF) and 0.6 mM *p*-methylphenylalanine (pMeF) from 1 mM of the corresponding aldehydes within 6 h with 5 mM L-Glu as the amino donor for TyrB. These results demonstrate that the designed pathway is efficient for synthesizing non-canonical amino acids (ncAAs).

Next, we assessed the biosynthesis of ncAAs in *E. coli* BL21 (PpLTA-RpTD) cultures, using 1 mM *p*-iodobenzaldehyde or *p*-methylbenzaldehyde, 50 mM glycine, and 20 μM pyridoxal-5′-phosphate (PLP) as substrates. The substrates were added during the mid-exponential phase, along with isopropyl $\beta$-D-thiogalactopyranoside (IPTG) to induce protein expression. However, *E. coli* BL21 (PpLTA-RpTD) was less efficient in producing pIF and pMeF from aldehydes in culture compared to the whole-cell catalysis experiment. Detailed analysis of intermediates and metabolites in the cultures revealed that the benzaldehydes were partially metabolized to benzyl alcohol, with only 0.56 mM pIF and 0.26 mM pMeF produced after 12 h (Fig. 2a and Supplementary Fig. 5).

To improve this, we expressed the two enzymes in *E. coli* MG1655 RARE (DE3)[43], a strain with six aldehyde reductase genes knocked out ($\Delta dkgB$, $\Delta yeaE$, $\Delta yqhC$-$dkgA$, $\Delta yahK$, $\Delta yjgB$), which is deficient in reducing aromatic aldehydes. This new strain, *E. coli* RARE (PpLTA-RpTD), produced 0.8 mM pIF after 12 h, and the formation of *p*-iodobenzyl alcohol was reduced by more than ten-fold (Fig. 2a). In contrast, the wild-type *E. coli* MG1655 (DE3) strain harboring PpLTA and RpTD produced similar quantities of *p*-iodobenzyl alcohol and pIF (0.48 mM) to *E. coli* BL21 (PpLTA-RpTD), suggesting that the increased pIF titer in the *E. coli* RARE (PpLTA-RpTD) strain was due to the knockout of the aldehyde reductase genes. Additionally, the titer of pMeF was improved to 0.44 mM in this strain (Supplementary Fig. 5). *p*-iodobenzaldehyde initially inhibited the growth of *E. coli* RARE (PpLTA-RpTD), as indicated by a reduction in OD$_{600}$. However, after 10 h, the OD$_{600}$ values began to recover (Supplementary Fig. 6), possibly due to the conversion of most of the aldehyde, leading to a diminished inhibitory effect. Furthermore, the growth of *E. coli* was minimally impacted by the intermediate metabolite in this biosynthetic pathway. (Supplementary Fig. 7).

Next, we attempted to couple the biosynthetic pathway in situ with GCE to produce superfolder GFP (sfGFP) with pIF. First, the *E. coli* RARE (DE3) strain was transformed with two plasmids (Fig. 2b) encoding a mutant of the widely used GCE module containing *Methanosarcina mazei* pyrrolysyl-tRNA synthetase and tRNA pair (pCDF-*Mm*pIFRS-tRNA$^{Pyl}_{CUA}$)[44], and a sfGFP mutant (pET22b-sfGFP$_{Y151TAG}$), site-specific incorporation of pIF. We then analyzed the expression levels of pET22b-sfGFP$_{Y151TAG}$ under different concentrations of pIF. Our results revealed that supplementing the culture medium with 1 mM pIF resulted in successful production of sfGFP$_{Y151pIF}$ (Supplementary Fig. 8). The *E. coli* RARE (DE3) strain was then transformed with a third plasmid (Fig. 2b) encoding the biosynthesis module (pACYCDuet-PpLTA-RpTD) to couple in vivo ncAA biosynthesis and genetic code expansion. The resulting strain was cultivated in the presence of substrates and cofactors. Green fluorescence was observed in the harvested cells of this strain, showing 68% relative fluorescence compared to cells grown with 1 mM pIF as a positive control.

In contrast, *E. coli* BL21 (DE3) expressing the same modules produced six-fold lower levels of sfGFP due to the lower pIF titer (Fig. 2c). Negative control experiments confirmed that both *p*-iodobenzaldehyde and glycine were essential substrates, while PLP was only marginally beneficial for both ncAA biosynthesis and sfGFP production (Fig. 2d).

To confirm that the biosynthesized pIF was incorporated, the sfGFP mutant with a C-term His$_6$ tag was purified in 27.3 mg L$^{-1}$ yield and subjected to Mass Spectrometry. The experimental spectrum with a peak at 27916 Da matches the calculated molecular weight of sfGFP

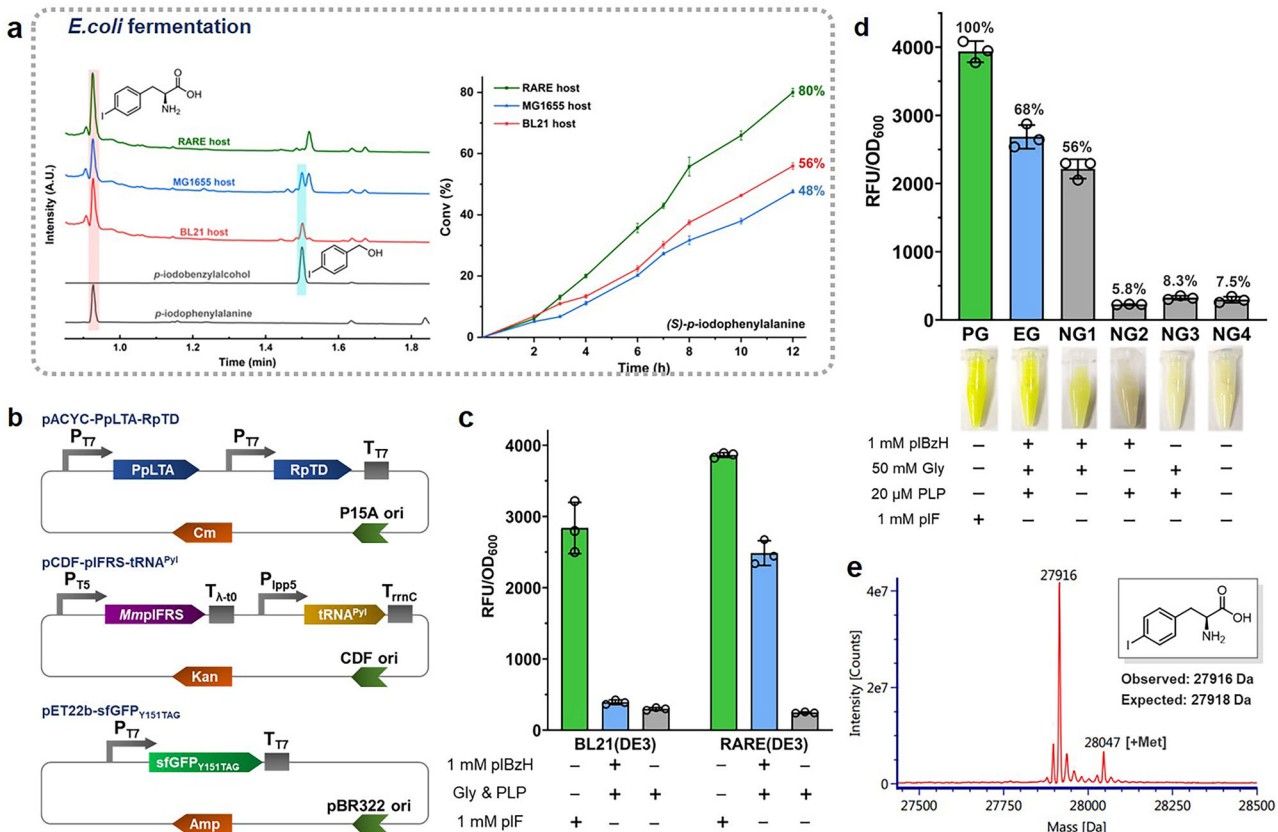

**Fig. 2 | Initial characterization of engineering *E. coli* strain for pIF synthesis and incorporation into sfGFP. a** Transformation of *p*-iodobenzaldehyde (pIBzH) by various *E. coli* strains with reconstituted biosynthetic pathways was evaluated. The reaction activity of different *E. coli* strains for the conversion of pIBzH to pIF was assessed by supplementing LB cultures with 1 mM pIBzH at the time of expression induction. The products were analyzed using UPLC-MS. Conversions within 12 h were calculated. Error bars represent the mean ± s.d. of $n = 3$ independent biological samples. **b** Depiction of different plasmids for pIF synthesis and incorporation into sfGFP. pACYCDuet-1 contained threonine aldolase (PpLTA) and threonine deaminase (RpTD) genes for cascade catalysis, pCDF plasmid was used to express the GCE system for pIF encoding, and pET22b contained sfGFP bearing an amber mutation at Y151. Genes were expressed under IPTG-induction, except that tRNA was controlled by a constitutive promoter. **c** Production of sfGFP using pIF, biosynthesized from pIBzH, was carried out in *E. coli* BL21 (DE3) and RARE (DE3) hosts. The efficiency of production was evaluated based on the fluorescence intensity of sfGFP. The positive group was in the presence of 1 mM pIF (green bar), the

experimental group was in the presence of 1 mM pIBzH (blue bar), and the negative group was without pIF or pIBzH (gray bar). Error bars represent the mean ± s.d. of $n = 3$ independent samples. **d** The effect of different reaction components on sfGFP production in *E. coli* RARE (DE3) carrying three plasmids was evaluated. In the negative control groups, one of the components was omitted, or no additional component was added, compared to the experimental group. Cultures were supplemented with 1 mM pIF or 1 mM pIBzH, and fluorescence was measured after 24 h of induction. The fluorescence intensity of cells cultured with 1 mM pIF was set as 100% for comparison. Error bars represent the mean ± s.d. of $n = 3$ independent samples. PG, positive group with 1 mM pIF; EG, experimental group with 1 mM pIBzH; NG1, negative group, PLP was omitted; NG2, negative group, Gly was omitted; NG3, negative group, pIBzH was omitted; NG4, negative group, without any components. **e** Mass characterization of sfGFP with pIF biosynthesized from pIBzH. The expected molecular mass (MW) value of sfGFP with pIF at Y151 was 27918 Da; observed MW value (as shown) was 27916 Da. Source data of (**a, c**) and (**d**) are provided in the Source Data file.

containing a pIF residue (Fig. 2e). This result confirmed that the biosynthesized pIF was incorporated into the target protein.

### Substrate scope of the biosynthetic pathway

Building on the successful biosynthesis of pIF and its incorporation into sfGFP as proof of concept, we expanded our investigation to synthesize a broader range of aromatic non-canonical amino acids (ncAAs), aiming to demonstrate the versatility of the engineered biosynthetic pathway. For this purpose, we first produced several ncAAs not commercially available using *E. coli* BL21 (PpLTA-RpTD) whole-cell catalysts and utilized them as analytical standards for quantification. By systematically optimizing reaction parameters, including substrate and co-substrate ratios (Supplementary Fig. 9), we successfully produced and characterized fifteen ncAAs with isolated yields ranging from 5.5% to 70.4% (Supplementary Fig. 10).

With both commercial and in-house synthesized ncAA standards available, we systematically assessed the versatility of the biosynthetic pathway using lyophilized *E. coli* RARE/pACYC-CsLTA-RpTD cells. The

whole-cell catalyst (10 mg/mL) efficiently converted forty aldehyde substrates containing diverse functional groups (Fig. 3a) into their corresponding ncAAs, achieving yields ranging from 20% to 98%, as confirmed by quantitative UPLC-MS analysis (Supplementary Fig. 11).

We then proceeded to test the aldehyde substrates for ncAA synthesis in *E. coli* RARE (DE3) strains harboring the biosynthetic module in culture. Four variants of LTA were screened to evaluate both the efficiency and substrate scope of ncAA biosynthesis (Supplementary Fig. 12). After twelve hours of incubation, the concentrations of ncAAs in cell cultures were quantified using UPLC-MS. Among the variants tested, *E. coli* RARE (CsLTA-RpTD) harboring CsLTA from *Cellulosilyticum* sp. I15G10I2, replacing PpLTA, exhibited the highest activity across most substrates. In total, twenty-one ncAAs were produced with yields exceeding 50% (Fig. 3b).

We found that substituents at the *meta*- (**6**−**16**) and *para*-positions (**17**−**32**) were better tolerated by the biosynthetic pathway than those at the *ortho*-position (**1**−**5**). Furthermore, bicyclic aromatic and heterocyclic substrates (**34**−**40**) were also converted into the

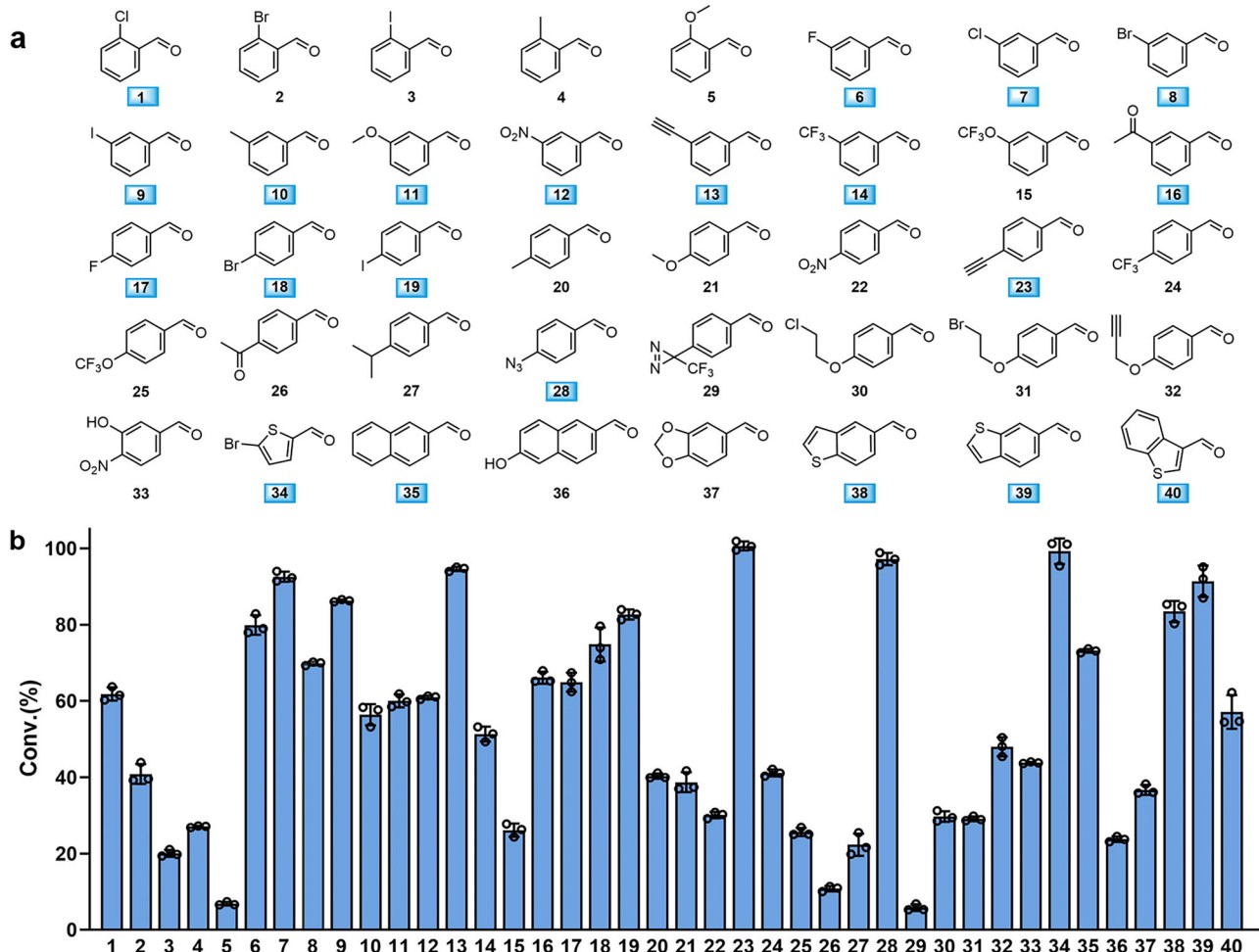

**Fig. 3 | Biosynthetic efficiency of ncAAs from aromatic aldehyde precursors in culture by *E. coli* RARE (CsLTA-RpTD). a** Structures of various aromatic aldehyde precursors used in this work. Substrates with a conversion rate higher than 50% were highlighted with blue boxes. **b** Conversion of different aromatic aldehyde precursors by *E. coli* RARE (CsLTA-RpTD). Aromatic aldehydes, Gly and PLP with a final concentration of 1 mM, 50 mM and 20 µM were added to the medium in 96-deep well plates at the same time with IPTG for inducing expression. Products were analyzed by UPLC-MS after reaction for 12 h at 30 °C. Error bars represent the mean ± s.d. of *n* = 3 independent samples. Source data of (**b**) are provided in the Source Data file.

corresponding ncAAs with moderate to high yields. Among these substrates, compounds **13, 16, 23, 26, 28**, and **32** contained bioorthogonal reactive groups suitable for site-specific bioconjugation reactions, while compound **21** featured functional groups relevant for studying protein post-translational modifications.

To investigate the stereochemical preferences and reaction rates of the pathway enzymes, we performed detailed kinetic analyses of PpLTA, CsLTA, and RpTD using *para*-(trifluoromethoxy)benzaldehyde as the substrate, enabled by the successful purification of diastereoisomeric phenylserine intermediates. Steady-state kinetic measurements revealed that PpLTA and CsLTA exhibited catalytic efficiencies ($k_{cat}/K_m$ of 34600 M$^{-1}$ s$^{-1}$ and 18206 M$^{-1}$ s$^{-1}$, respectively) for the production of (*2S,3R*)-*p*-(trifluoromethoxy)phenylserine (Supplementary Fig. 13). In contrast, the $k_{cat}/K_m$ value for the generation of the (*2S,3S*)-diastereomer were significantly lower: 3621 M$^{-1}$ s$^{-1}$ (PpLTA)and 13564 M$^{-1}$ s$^{-1}$ (CsLTA). Subsequent analysis of RpTD demonstrated a similar stereodivergence: the enzyme efficiently processed the (*2S,3R*)-phenylserine intermediate to the keto-acid ($k_{cat}/K_m$ = 65231 M$^{-1}$ s$^{-1}$, Supplementary Fig. 14) but showed reduced activity toward the (*2S,3S*)-isomer ($k_{cat}/K_m$ = 405 M$^{-1}$ s$^{-1}$). Based on these kinetic results, we concluded that the (*2S,3S*)-phenylserine intermediate is critical for pathway flux.

To identify the native aminotransferase responsible for the final step of ncAA biosynthesis, we assayed the activity of purified *E. coli* aminotransferases against representative keto-acid intermediates. Screening revealed that aromatic amino acid aminotransferase (TyrB)[41], aspartate aminotransferase (AspC)[45], and branched chain amino acids aminotransferase (ilvE)[46] exhibited promiscuous activity toward the substrates. While TyrB and AspC showed robust catalytic efficiency, IlvE displayed significantly lower activity (Supplementary Fig. 15), suggesting that TyrB and AspC are the primary contributors to the transamination step.

Next, we investigated the rate-limiting step for substrates that exhibited low conversion in the culture. UPLC analysis, using substrate and intermediate standards, revealed that the transaminase was responsible for the low conversions of *ortho*-substituted substrates, leading to the accumulation of α-keto acids (Supplementary Fig. 16). Conversely, LTA was the limiting factor for substrates with *para*-substituents, resulting in the residue of starting materials, with no detectable α-keto acids (Supplementary Fig. 17). These observations illustrate that the synthetic efficiency of this biosynthetic pathway could be further improved through protein engineering of the key enzymes tailored for specific ncAAs of interest.

## Streamlined production of sfGFP coupling biosynthesis pathway with various OTS

We next aimed to evaluate the general compatibility of ncAA production with different OTSs. To achieve this, we constructed mutants of the *Methanosarcina mazei* pyrrolysyl-tRNA synthetase (*Mm*PylRS) and its cognate tRNA[47–49], as well as a mutant of the *Methanosarcina barkeri* pyrrolysyl-tRNA synthetase (*Mb*PylRS) and its cognate tRNA[50], both inserted into the pCDF vector. Additionally, a mutant of the *Methanococcus janaschii* tyrosyl-tRNA synthetase and its corresponding tRNA (*Mj*TyrRS/*Mj*tRNA$^{Tyr}_{CUA}$)[51–54] was introduced into the pUltra vector (Fig. 4a). To assess the performance of these OTSs, we expressed the sfGFP$_{Y151TAG}$ as reporter protein. Initially, the LTAs were screened for sfGFP$_{Y151pIF}$ production to evaluate the efficiency of ncAA biosynthesis and incorporation. The results showed that sfGFP production (Fig. 4b) correlated well with the ncAA titer (Supplementary Fig. 18, substrate **19** for pIF). Notably, the strain containing CsLTA produced 83% sfGFP-based fluorescence to the positive control. Consequently, CsLTA was selected for the following experiments.

We then investigated the biosynthesis and incorporation of nineteen ncAAs, titers of which were over 500 µM, by coupling the biosynthetic pathway and OTS. As clearly shown in Fig. 4c, the biosynthetic pathway is highly efficient in cooperation with all three OTSs. We selected the 15 most fluorescent strains to isolate and quantify the recombinant sfGFP. The isolated yields ranged from 12 to 71 mg per liter of culture after purification. Nearly half of these sfGFP with ncAAs reached more than 50% of the yield of wild type sfGFP (Supplementary Table 1 and Fig. 4d). Mass spectrometry confirmed the incorporation of the ncAAs in sfGFPs (Fig. 4e and Supplementary Fig. 19). The results indicated that the biosynthetic pathway was compatible with the tested OTSs, although the growth inhibition by aldehyde substrates needs to be tested case-by-case (Supplementary Fig. 20). Most of the modified sfGFPs showed comparable or greater normalized fluorescence than that observed in the positive control groups, which were cultured in the presence of the corresponding ncAAs (1 mM final concentration). The low observed fluorescence of the *o*-bromosubstrate relative to that of the control group could be attributed to the low conversion of the substrate to ncAA (40%, Fig. 3b, substrate **4**). On the other hand, the low fluorescence intensity of *p*-bromophenylalanine (pBrF) in both the experimental and positive control groups may be due to poor incorporation rate by the *Mm*PylRS mutant.

The advantage of coupling ncAAs biosynthesis and GCE is revealed by the 2 times greater normalized fluorescence with *meta*-nitro-substituted substrate and *para*-borono-substituted substrate than that observed in the control groups with *meta*-nitrophenylalanine (mNF) and *para*-boronophenylalanine (pBoF). More importantly, proteins containing these two residues had great potential in applications. Nitrated proteins may be useful for increasing protein immunogenicity[55]. pBoF has been applied in bioconjugation and bioorthogonal labeling of proteins, as well as in enzymatic catalysis[56,57]. ncAAs containing bioorthogonal reactive groups for site-specific bioconjugation reactions, such as *meta*-acetylphenylalanine (mAcF), *para*-ethynylphenylalanine (pENF), *para*-azidephenylalanine (pAzF) and *para*-O-propargylphenylalanine (pPrF), were successfully synthesized and incorporated to deliver sfGFP in yields ranging from 31 to 71 mg per liter of culture. All of these examples clearly verified the excellent efficiency, flexibility and broad compatibility of the biosynthetic pathway for ncAAs synthesis and protein production with GCE.

## Biosynthesis of macrocyclic peptides with ncAAs

With a robust platform in hand, we aimed to demonstrate its application in the production of various polymers containing ncAAs. Macrocyclic peptides, which possess unique pharmacological properties distinct from other well-established therapeutic molecular classes, have garnered significant attention from the scientific community in recent years[58]. We then explored the capability of our platform in conjunction with the split-intein circular ligation of peptides and proteins (SICLOPPS) system to produce macrocyclic peptides[59]. The SICLOPPS system has recently been shown to be compatible with genetic code expansion, allowing for the incorporation of ncAAs[60].

To achieve this, we combined our platform for ribosomal ncAA incorporation with SICLOPPS to create unnatural cyclic peptides (Fig. 5). First, we validated the activity of the synthetic *Npu* sequence, confirming the generation of cyclo-CLLFVY using UPLC-MS (Supplementary Fig. 21). Initial experiments showed low fluorescence intensity in sfGFP when using a single-copy pAzFRS-tRNA system in the pUltra vector for pAzF incorporation. To address this, we employed arabinose-inducible promoters in the pEVOL vector to drive the expression of two pAzFRS copies, which significantly improved the production of pAzF-containing proteins (Supplementary Fig. 22). The improved performance was further validated in the production of pAzF-containing cyclic peptides, as evidenced by both Fig. 5b and Supplementary Fig. 23. Next, we produced derivatives of cyclo-CLLFVY by incorporating four different ncAAs at three distinct positions. The peptides were detected in all experimental samples (EG) and in the positive control group supplemented with the corresponding ncAAs (PG), but were absent in the negative control group that lacked the substrates (NG). The intensities of most peptides were within a threefold range of that of cyclo-CLLFVY. It was evident that peptide productivity was influenced by the biosynthetic pathway, the OTSs used, and the specific incorporation site of the ncAA. Notably, the coupled system demonstrated exceptional efficiency in incorporating pAzF, resulting in a 25-fold higher intensity than that observed when pAzF was directly supplemented. The intensity of peptides containing pENF was also higher than that of cyclo-CLLFVY. However, the system was less efficient in producing peptides containing pBoF and pPrF.

## Biosynthesis of antibody fragments with ncAAs in *E. coli*

We next explored the feasibility of coupling this biosynthetic pathway with the production of antibody fragments incorporating ncAAs. To this end, we established a platform for the recombinant expression of three antibody fragments: an anti-HER2 single-chain fragment variable (scFv)[61], an anti-HER2 antigen-binding fragment (Fab)[35], and a J591-Fab[62]. The anti-HER2 antibody is clinically used for treating HER2-overexpressing breast cancer, while the J591 antibody targets prostate-specific membrane antigen for Near-IR fluorescence imaging of prostate cancer. For the incorporation of ncAAs, we selected S9 and K42 of the anti-HER2 scFv, A121 of the anti-HER2 Fab Heavy chain and A116 of the J591 Fab Heavy chain. *p*-azidophenylalanine was chosen for incorporation by the pAzFRS (2×)/tRNA pair system, owing to its unique reactivity in bioorthogonal chemistry.

The antibody mutants were recombinantly expressed in *E. coli* RARE (DE3), which was transformed with three plasmids (Fig. 6a), in culture medium supplemented with 1 mM *p*-azidobenzaldehyde. The yields of the expressed anti-HER2 scFv mutants, S9pAzF and K42pAzF, were 1.5 mg L$^{-1}$ and 1.44 mg L$^{-1}$, respectively, compared to 3.2 mg L$^{-1}$ for the wild-type protein (Fig. 6b and Supplementary Table 3). For the anti-HER2 Fab mutants, the yield of A121pAzF was 1.53 mg L$^{-1}$, while the wild-type yield was 2.82 mg L$^{-1}$. The A116pAzF mutant of J591-Fab yielded 2.93 mg L$^{-1}$, compared to 5.25 mg L$^{-1}$ for the wild-type Fab. Incorporation of pAzF into all mutants was confirmed by high-resolution mass spectrometry.

To confirm the bioactivity of the engineered antibody fragments, we site-specifically conjugated anti-HER2-Fab-A121pAzF with the fluorescent probe AF488-DBCO (Supplementary Fig. 24). Confocal imaging revealed specific and robust staining of HER2-positive cells by the labeled Fab fragment (Fig. 6d), validating its target-binding capability. These results confirm that our streamlined platform reliably produces functional ncAA-incorporated antibody fragments, positioning them as promising candidates for both diagnostic imaging and therapeutic development.

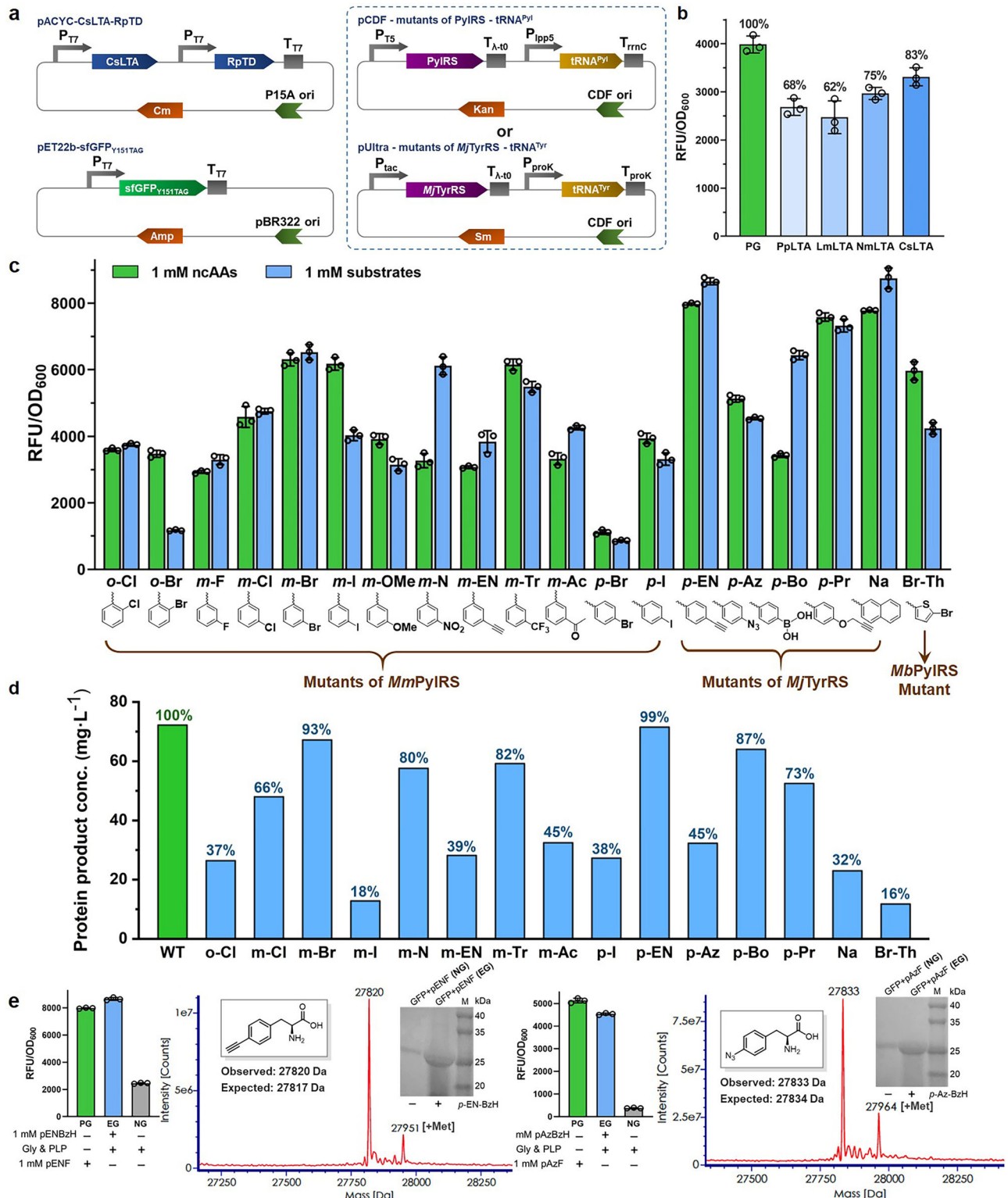

## Discussion

Aromatic amino acids, including phenylalanine derivatives, are among the most commonly used ncAAs in GCE technology for a variety of applications. While many aromatic amino acids are commercially available or can be synthesized, the high cost and limited availability of these compounds remain significant barriers to the widespread application of GCE technology. The biosynthetic pathway and platform we present here offer a practical solution to this challenge. The key advantage is that the cost of the starting materials, aryl aldehydes, are

the precursors in the pathway, is orders of magnitude lower than that of the corresponding ncAAs, while the productivity of proteins and peptides is comparable to or even higher than when using ncAAs. Additionally, the aldehyde substrates can easily penetrate the cell membrane, overcoming the permeability issues typically associated with ncAAs. Although the current biosynthetic pathway does not yet achieve high titers of all ncAAs for efficient GCE, its broad applicability provides a strong foundation for future engineering and optimization efforts to improve production efficiency.

**Fig. 4 | The ncAAs biosynthetic pathway was scalable and efficient for coupling with various orthogonal aaRS/tRNA. a** Plasmid design for coupling ncAA synthesis and incorporation into sfGFP. The plasmids used for ncAA synthesis and incorporation into sfGFP were the same as those described in Fig. 2b. These included plasmids encoding CsLTA and RpTD for ncAA synthesis, and plasmids encoding sfGFP$_{Y151TAG}$ as the reporter protein. Mutants of two types of PylRS/tRNA pairs were constructed in the pCDF vector, while mutants of the *Mj*TyrRS/tRNA pair were constructed in the pUltra vector. tRNA expression was driven by a constitutive promoter, whereas other genes were expressed under IPTG induction. **b** Detection of sfGFP$_{Y151pIF}$ fluorescence intensity in strains expressing different LTAs. Fluorescence intensity of sfGFP$_{Y151pIF}$ in strains expressing different LTAs was measured after 24 h of incubation. The fluorescence intensity of cells cultured with 1 mM pIF was set as 100%. For the experimental group, 1 mM pIBzH, 50 mM glycine, and 20 μM PLP were added to the culture at the same time as IPTG induction. Error bars represent the mean ± s.d. of $n = 3$ independent samples. **c** Fluorescence intensity of sfGFP containing different ncAAs. The fluorescence intensity of sfGFP$_{Y151TAG}$ was

measured in *E. coli* RARE (CsLTA-RpTD) strains containing various OTSs, after 24 h of fermentation. The culture was supplemented with either 1 mM ncAAs or 1 mM aromatic aldehyde substrates. Error bars represent the mean ± s.d. of $n = 3$ independent samples. **d** Yields of sfGFP containing ncAAs after purification. The yield of sfGFP containing ncAAs was determined after purification. To the culture, 1 mM aromatic aldehydes were added, and after 24 h of induction, the proteins were purified using a Ni-NTA column. The yield of wild-type sfGFP was set as 100% for comparison. **e** Mass spectra of purified sfGFP with *p*-ethynylphenylalanine (pENF) or *p*-azidephenylalanine (pAzF) and corresponding detection of fluorescence intensity. For fluorescence intensity, error bars represent the mean ± s.d. of $n = 3$ independent samples. Mass spectra of purified sfGFP incorporated with other ncAAs were shown in Supplementary Fig. 19. PG, positive group with 1 mM ncAAs; EG, experimental group with 1 mM aromatic aldehydes; NG, negative group without ncAAs or aromatic aldehydes. Source data of (**b–e**) are provided in the Source Data file.

Looking ahead, several research directions are worth pursuing. First, optimizing the biosynthetic pathway to efficiently produce a wider range of ncAAs, including both aromatic and aliphatic types, tailored to specific applications, should be a priority. Secondly, aldehydes are not ideal materials due to their reactivity and toxicity. To address this, extending the biosynthetic pathway using enzymes such as transaminase, alcohol dehydrogenase, or carboxylic acid reductases could provide a consistently low concentration of aldehyde from their corresponding precursors, amines, alcohols, or carboxylic acids. Third, exploring the simultaneous synthesis of multiple ncAAs, followed by the integration of several orthogonal translation systems, such as quadruplet codons, within a single cell, would be highly valuable. Finally, it would be worthwhile to evaluate the feasibility of assembling and implementing this coupled system in eukaryotic organisms, such as *Pichia pastoris*.

In this work, we integrate the biosynthesis of aromatic amino acids with genetic code expansion in *E. coli*, enabling the production of proteins and peptides containing incorporated ncAAs. The platform efficiently produces a diverse range of high-value ncAAs, with important applications in catalysis and bioorthogonal chemistry. We envision that this platform will greatly enhance the programmable production of peptides, enzymes, and antibody fragments incorporating ncAAs, thereby accelerating their adoption in a variety of applications.

## Methods
### Strains and chemicals
Genes were codon-optimized, synthesized, and cloned into pET28a or pET22b expression plasmids by Genewiz. Amino acid and nucleotide sequences of proteins were provided in Supplementary data 1. The primers used in this study for cloning were synthesized by Genewiz (Listed in Supplementary data 2). Competent *E. coli* BL21 (DE3) cells were obtained from Sangon Biotech, while *E. coli* MG1655 (DE3) cells were purchased from Zomanbio. SK-Br-3 and MDA-MB-468 cells were from the Cell Bank of the Chinese Academy of Sciences (Shanghai, China). The expression strain *E. coli* RARE (DE3) was acquired from Addgene and transformed into competent cells using the Super Receptive Cell Preparation Kit from Sangon Biotech. Antibiotics and media components were sourced from Sangon Biotech as well. Homologous recombinase and T4 ligase were purchased from Vazyme, and all restriction endonucleases were obtained from Takara. Commercial substrates were sourced from Bidepharm and used without further purification. Deuterated solvents were purchased from Energy Chemical. The high-pressure homogenizer and Ni-NTA Union 6FF agarose were obtained from Union Biotech.

### Quantification of ncAAs and other compounds in culture
Chemically competent *E. coli* RARE (DE3) cells were transformed with the plasmid pACYC-CsLTA-RpTD. Single colonies were selected and

inoculated into LB medium containing 35 μg mL$^{-1}$ chloramphenicol (Cm). After overnight incubation, the cultures were transferred to fresh LB medium (1% inoculum) containing 35 μg mL$^{-1}$ Cm and grown at 37 °C with shaking at 220 rpm until the OD$_{600}$ reached 0.8–1.0. The temperature was then lowered to 30 °C, and protein expression was induced by adding IPTG to a final concentration of 1 mM. The reaction mixture was added to the medium to reach a final concentration at 1 mM aromatic aldehyde substrates, 50 mM glycine (Gly), and 20 μM pyridoxal 5′-phosphate (PLP) at the same time with expression induction, and the culture was continued for 24 h at 30 °C with shaking at 230 rpm. The concentrations of ncAAs, substrates, and keto acid intermediates in the cell cultures were measured by UPLC-UV or UPLC-MS analysis and quantified using standard curves.

### General procedure for expression and fluorescence measurement of sfGFP with ncAAs
Plasmid pET22b-sfGFP(Y151TAG) was introduced into *E. coli* RARE (DE3), and the strain was then prepared as competent cells. Co-expressing plasmid pACYC-CsLTA-RpTD and plasmids for expressing different aaRS/tRNA pairs were separately co-transformed into *E. coli* RARE (DE3)/pET22b-sfGFP(Y151TAG) competent cells. When pCDF was used as the aaRS/tRNA vector, cells were plated on agar containing 35 μg mL$^{-1}$ chloramphenicol (Cm), 100 μg mL$^{-1}$ ampicillin (Amp), and 50 μg mL$^{-1}$ kanamycin (Kan). When pUltra was used as the aaRS/tRNA vector, cells were plated on agar containing 35 μg mL$^{-1}$ Cm, 100 μg mL$^{-1}$ Amp, and 100 μg mL$^{-1}$ streptomycin (Sm). Thus, the three-plasmid co-expression strains were successfully generated. Single colonies were selected and inoculated into LB medium containing the appropriate antibiotics. After overnight culture, the cells were transferred to fresh LB medium (1% inoculum) containing the corresponding antibiotics and cultured at 37 °C with shaking at 220 rpm until the OD$_{600}$ reached 0.8-1.0. IPTG was then added to a final concentration of 1 mM. The reaction mixture was added to the medium to reach a final concentration of 1 mM aromatic aldehyde substrates, 50 mM glycine (Gly), and 20 μM pyridoxal 5′-phosphate (PLP), and the culture was continued for 24 h at 30 °C with shaking at 230 rpm. 1 mL of *E. coli* culture was collected, and the bacterial concentration was measured using a UV spectrophotometer. The cells were harvested by centrifugation, washed twice with PBS (pH 7.5), and then suspended in 1 mL of PBS (pH 7.5). Fluorescence intensity (Ex: 485 nm; Em: 528 nm) was measured using 200 μL of the suspended cells in a plate reader. The ratio of fluorescence intensity to bacterial concentration (RFU/OD$_{600}$) was used as an indicator of sfGFP expression intensity in the recombinant strains. A culture supplemented only with glycine (Gly) and pyridoxal 5′-phosphate (PLP), but without substrates, was used as the negative control, while a culture supplemented with 1 mM aromatic non-canonical amino acids (ncAAs) served as the positive control.

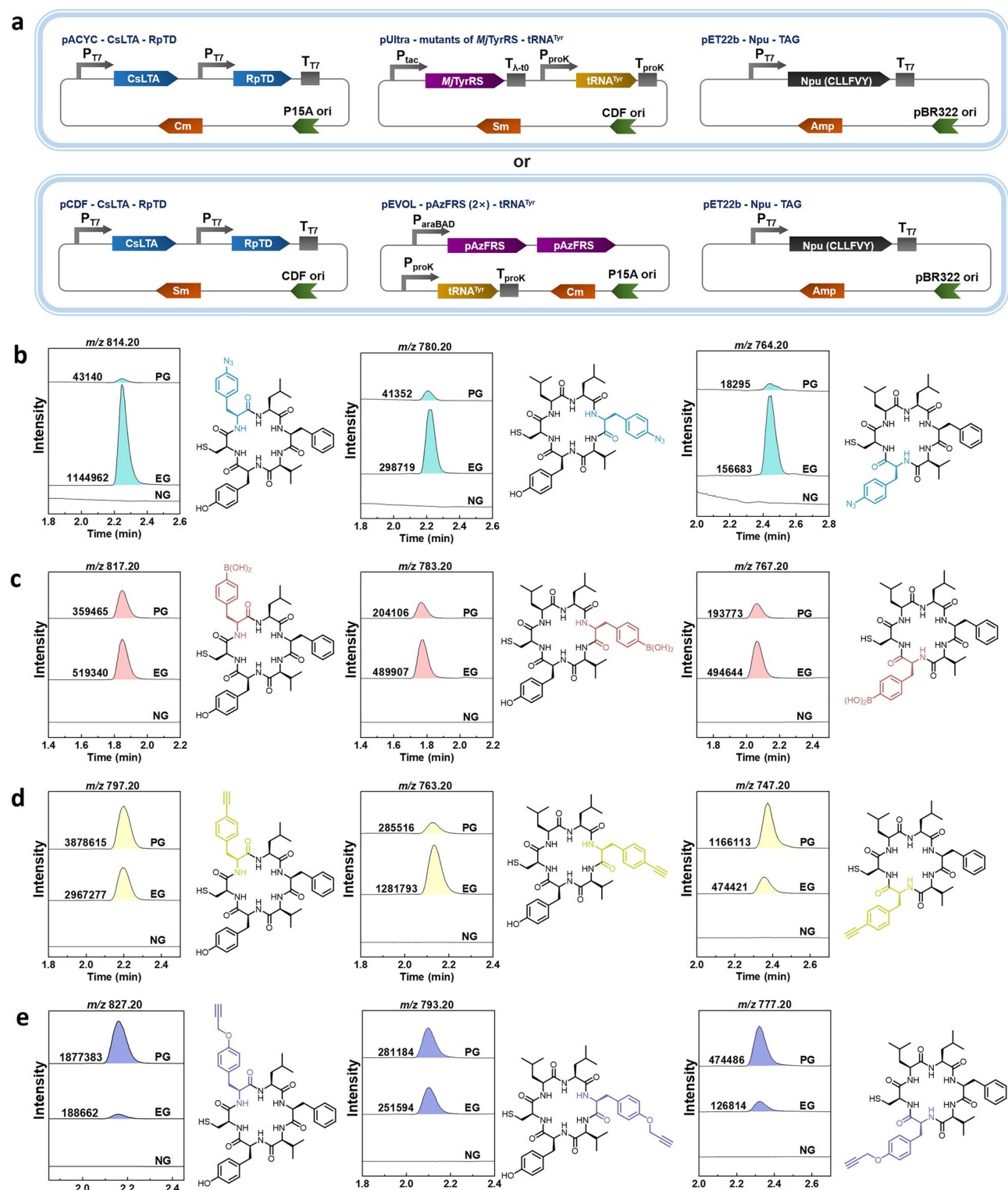

**Fig. 5 | The ncAAs biosynthetic pathway applied for intracellular synthesis of peptide macrocycles. a** Plasmid design for coupling ncAAs biosynthesis and incorporation into *Npu* of cyclo-CLLFVY. The plasmids used for the expression of CsLTA and RpTD for ncAA biosynthesis were the same as those shown in Fig. 2b, paired with mutants of the *Mj*TyrRS/tRNA pair as described in Fig. 4a. CsLTA and RpTD were cloned into the pCDFDuet-1 vector and paired with the pAzFRS (2×)/ tRNA pairs, which were constructed in the pEVOL vector. The expression of the pAzFRS (2×) was controlled by the arabinose-inducible pBAD promoter, while tRNA expression was driven by a constitutive promoter. Other genes were expressed

under IPTG induction. **b** Cyclo-CLLFVY, with pAzF at positions 2, 4, or 6, was incorporated by the pAzFRS (2×)/tRNA$^{Tyr}_{CUA}$ pair. **c** Cyclo-CLLFVY, with pBoF at positions 2, 4, or 6, was decoded by the pBoFRS/tRNA$^{Tyr}_{CUA}$ pair. **d** Cyclo-CLLFVY, with pENF at positions 2, 4, or 6, was incorporated by the pCNFRS/tRNA$^{Tyr}_{CUA}$ pair. **e** Cyclo-CLLFVY, with pPrF at positions 2, 4, or 6, was incorporated by the pCNFRS/ tRNA$^{Tyr}_{CUA}$ pair. The detail of Mass Spectrometry (MS) of macrocycles was listed in Supplementary Table 2. PG, positive group with 1 mM ncAAs; EG, experimental group with 1 mM aromatic aldehydes; NG, negative group without ncAAs or aromatic aldehydes. Source data of **b**–**e** are provided in the Source Data file.

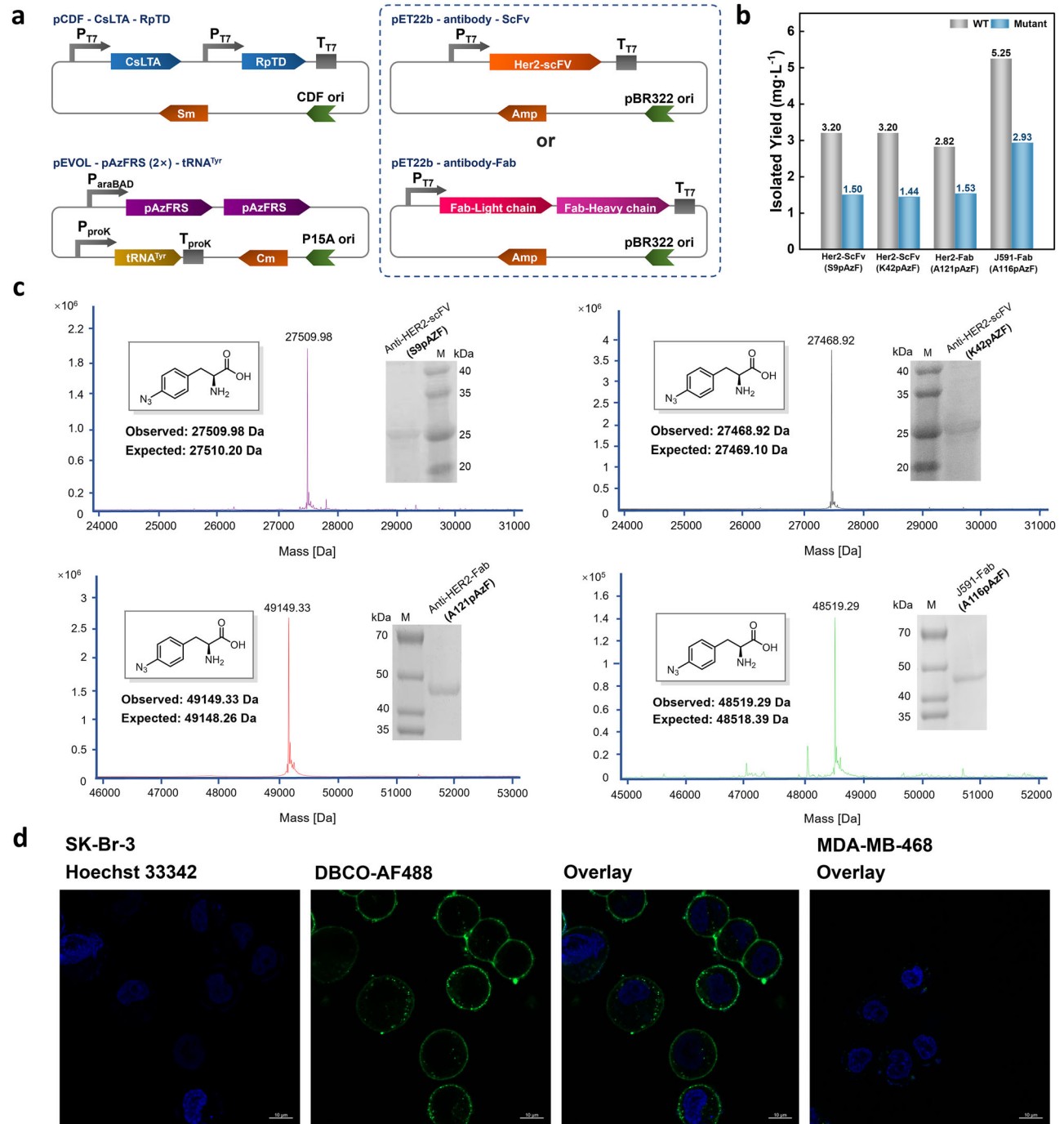

**Fig. 6 | The ncAAs biosynthetic pathway applied for intracellular synthesis of antibody fragments. a** Depiction of plasmids for ncAAs synthesis and incorporation into antibody fragments. Plasmids that expression of CsLTA and RpTD for ncAAs cascade catalysis and pAzFRS (2×)/tRNA^Tyr pairs were same as Fig. 5a. Mutants of four antibody fragments were constructed in the pET22b vector. pAzFRS (2×) was controlled by arabinose-controlled pBAD promoter, tRNA was controlled by constitutive promoter and other genes were expressed under IPTG-induction. **b** The yields of antibody fragments containing ncAAs (Mutant) after purification and comparison with wild type (WT). The cell was cultured with 1 mM pAzBzH, 50 mM Gly and 20 μM PLP and induced by arabinose and IPTG for 24 h. The proteins were purified with Ni-NTA column. **c** High-resolution mass of purified antibody fragments with pAzF (The high-resolution mass spectrometry of wild type antibody fragments were shown in Supplementary Fig. 25). The purification experiments were repeated one time with result. **d** Confocal images of Her2-positive SK-Br-3 cells and Her2-negative MDA-MB-468 cells stained with anti-Her2-Fab-A121pAzF-AF488 conjugate. In consistent with literature (Ref.35), Her2-negative MDA-MB-468 cells was exploited as negative control in the fluorescence assay rather than wild type antibody because the click reaction between azide and alkyne group is commonly known as classic biortho-gonal chemistry. Scale bars = 10 μm. Assays were repeated one time with result. Source data of (**b**) are provided in Source Data file.

## Purification of sfGFP with ncAAs

A single colony of the tri-plasmid strain was cultured overnight in LB medium containing the three appropriate antibiotics, then transferred to 500 mL of LB medium with the corresponding antibiotics at a 1% inoculum. Strain culture and sfGFP expression were carried out according to the previously described methods. The bacteria were harvested by centrifugation at 4 °C, 10,000 rpm for 10 min. The cell pellet was suspended in lysis buffer (50 mM Tris, pH 8.0, 250 mM NaCl) and lysed using a high-pressure homogenizer. The lysate was centrifuged at 4 °C, 18,000 rpm for 30 min, and the supernatant was

passed through a Ni-NTA agarose purification column pre-equilibrated with lysis buffer. The column was washed with 10 column volumes of washing buffer (50 mM Tris, pH 8.0, 250 mM NaCl, 25 mM imidazole), and the protein was eluted with 5 column volumes of elution buffer (50 mM Tris, pH 8.0, 250 mM NaCl, 250 mM imidazole) to obtain the target sfGFP. The protein solution containing high concentrations of imidazole was dialyzed against dialysis buffer (50 mM $Na_2HPO_4$, pH 7.5, 50 mM NaCl). After dialysis, the protein solution was rapidly frozen in liquid nitrogen and stored at −80 °C. The identities of the purified proteins were confirmed by SDS-PAGE and high-resolution mass spectrometry (MS).

### Macrocyclic peptide biosynthesis and analysis

Plasmid encoding pET22b-Npu with the TAG mutant, plasmid encoding pACYC-CsLTA-RpTD or pCDF-CsLTA-RpTD, and plasmids for expressing various aaRS/tRNA pairs were co-transformed into *E. coli* RARE (DE3) competent cells. The strains were screened on selective agar plates containing 35 µg mL$^{-1}$ chloramphenicol (Cm), 100 µg mL$^{-1}$ ampicillin (Amp), and 100 µg mL$^{-1}$ streptomycin (Sm) to obtain engineered strains capable of site-specific incorporation of biosynthetic non-canonical amino acids (ncAAs) into the desired macrocycle via *Npu* intein. Single colonies were selected and cultured overnight in LB medium containing the three appropriate antibiotics, then inoculated at a 1% rate into fresh LB medium until the $OD_{600}$ reached 0.8–1.0. The temperature was then lowered to 30 °C, IPTG was added to a final concentration of 1 mM, and ∟-arabinose was added to a final concentration of 0.2% to induce protein expression. For the positive control group, ncAAs were added to a final concentration of 1 mM. For the experimental group, the reaction mixture was added to the medium at final concentrations of 1 mM aromatic aldehyde substrates, 50 mM glycine (Gly), and 20 µM pyridoxal 5'-phosphate (PLP). All cultures were then shaking at 230 rpm for 24 h. After harvesting by centrifugation, the cell pellet was suspended in 1% acetonitrile and sonicated for 30 min. The lysate was centrifuged at 13,000 rpm for 5 min, and the supernatant was filtered. Analysis of the macrocyclic peptides was performed using LC-MS with single ion recording (SIR) in positive mode.

### Expression and purification of pAzF-containing antibody fragments

To enhance the expression of the target protein, the commercial plasmid pEVOL-pAzFRS-tRNA$^{Tyr}_{CUA}$, which contains a double-copy pAzFRS gene, was selected for pAzF insertion. This plasmid includes the P15A replicative origin and a chloramphenicol (Cm) resistance marker. The CsLTA and RpTD co-expression gene cassette was cloned into the pCDFDuet-1 vector, which carries the CDF replicative origin and a streptomycin (Sm) resistance marker, using Gibson assembly. Different expression genes encoding an antibody fragment with the TAG mutation were cloned to pET22b and co-transformed with pCDF-CsLTA-RpTD and pEVOL-pAzFRS-tRNA$^{Tyr}_{CUA}$ into *E. coli* RARE (DE3) cells. Selection was performed on agar plates containing 35 µg mL$^{-1}$ Cm, 100 µg mL$^{-1}$ ampicillin (Amp), and 100 µg mL$^{-1}$ Sm to obtain engineered strains capable of site-specific incorporation of biosynthetic pAzF into recombinant proteins. Single colonies were selected and cultured overnight in LB medium with the three antibiotics, then inoculated at a 1% rate into fresh medium with the three antibiotics. When the $OD_{600}$ reached 0.8-1.0, IPTG was added to a final concentration of 1 mM, and ∟-arabinose was added to a final concentration of 0.2% to induce protein expression. The temperature was lowered to 30 °C, and the reaction mixture was then added to the medium at a final concentration of 1 mM *p*-azide-benzaldehyde, 50 mM glycine (Gly), and 20 µM pyridoxal 5'-phosphate (PLP). The cells were incubated for an additional 24 h at 30 °C with shaking at 230 rpm. After harvesting by centrifugation, the

proteins were purified using a Ni-NTA agarose purification column following the sfGFP purification protocol.

### Reporting summary
Further information on research design is available in the Nature Portfolio Reporting Summary linked to this article.

## Data availability
All data supporting the findings of this study are available within the paper and its Supplementary Information files, as well as from corresponding author upon request. Amino acid and nucleotide sequence of proteins were provided in Supplementary data 1. Primers used in this study for cloning were listed in Supplementary data 2. Source data are provided with this paper.

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

## Acknowledgments

We appreciate the assistance of staff from the Institutional Technology Service Center of Shanghai Institute of Materia Medica for their technical support. We gratefully acknowledge the National Natural Science Foundation of China (NSFC, No. 22071259 to C. L.; No. 22307126 to Y.X.), the Swiss National Science Foundation (SNSF, No. 206039 to F.P.S.), the Strategic Priority Research Program of the Chinese Academy of Sciences, (Grant No. XDB1060000 to C. L.); Shanghai Institute of Materia Medica (No. SIMM0220233001 to X.-H.C.), Chinese Academy of Sciences for their financial support. We are grateful to Prof. Jifeng Yuan (Xiamen University) for the helpful support on the host bacteria and the members of the Liao group for helpful discussions during the project.

## Author contributions

C.L. and X.-H.C. conceived and directed the project. J.Z., K.Y. and Y.X. performed most of the experiments and visualized the data. W.Z. and Y.L. assisted in experiments, data analysis and visualization. F.P.S. and Y.W. assisted in data analysis and supervision. J.Z., K.Y. and C.L. wrote the manuscript with input from all authors.

## Competing interests

Jingxuan Zhang and Cangsong Liao has submitted a patent application (application number: CN202411410965.8) based on this technology through Shanghai Institute of Materia Medica, Chinese Academy of Sciences. The other authors declare no competing interests.
