## [Peer Review File · Nature Communications]

A robust platform streamlining aromatic noncanonical amino acid biosynthesis and genetic code expansion in *Escherichia coli*

Corresponding Author: Dr Cangsong Liao

Version 0:

Reviewer comments:

Reviewer #1

(Remarks to the Author)

The study by Zhang et al. presents a robust platform for the biosynthesis of aromatic ncAAs and their incorporation into proteins via GCE in *Escherichia coli*, with the goal of enhancing large-scale protein production and expanding protein functionality. The authors successfully demonstrated the biosynthesis and incorporation of 40 distinct aromatic ncAAs, including para-azidophenylalanine (pAzF), para-boronophenylalanine (pBoF), meta-nitrophenylalanine (mNF), para-ethynylphenylalanine (pENF), para-O-propargylphenylalanine (pPrF), and various substituted phenylalanines. This work builds on the well-established concept of coupling orthogonal translation systems (OTSs) with engineered metabolic pathways in *E. coli* to enable efficient in vivo incorporation of ncAAs into target proteins.

The core novelty of this study lies in the application of metabolic engineering to develop a biosynthetic pathway that couples the in-situ production of ncAAs with genetic code expansion. This is achieved through a multi-enzyme cascade that converts aromatic aldehydes into ncAAs, which are then incorporated into modified peptides, enzymes, and antibodies. Specifically, the authors engineered the shikimate pathway - a natural pathway found in microbes, fungi, plants, and protists - to create a multi-enzyme cascade. This pathway typically converts simple carbohydrate precursors, such as phosphoenolpyruvate (PEP) and erythrose-4-phosphate (E4P), into chorismate, a key intermediate for the biosynthesis of aromatic amino acids. The key innovation here is the use of various aromatic substitutions of aryl-aldehydes, which are well tolerated by the system, to generate a library of substituted phenylalanine analogues. These analogues are subsequently incorporated into model proteins, demonstrating the versatility and potential of this approach.

As I understand it, their engineering approach includes three key components: (i) Enhancing endogenous enzymes from the shikimate and phenylalanine biosynthesis pathways; (ii) Introducing heterologous enzymes to extend substrate specificity and improve conversion efficiency; and (iii) Optimizing precursor availability to increase ncAA yields, using *Escherichia coli* as the expression platform. The feasibility of this approach was demonstrated through its application to various model proteins, including Superfolder GFP (sfGFP), macrocyclic peptides, and antibodies (such as the anti-HER2 single-chain variable fragment (scFv), anti-HER2 antigen-binding fragment (Fab), and J591-Fab). Undoubtedly, this strategy significantly simplifies the use of bioorthogonal labeling (bioconjugation) in protein science and protein-based material science, offering a powerful tool for advancing these fields.

As far as I can assess, the uptake of aryl-aldehydes is not without challenges: the efficiency of uptake is a critical factor determining the overall success of the system. The authors tested forty aldehyde analogues in *E. coli* RARE (DE3) strains harboring the biosynthetic module. However, it remains unclear when the co-expression of transport-enhancing genes is necessary and when passive diffusion alone suffices. The manuscript data suggest that intracellular accumulation of ncAAs varies depending on the aryl-aldehyde substrate and its metabolic conversion efficiency. Are there any general rules governing this process, or must it be determined on a case-by-case basis? Additionally, what intracellular ncAA concentrations (in terms of order of magnitude) are sufficient for efficient incorporation? Could the conversion of aryl-aldehydes to ncAAs produce intermediates that interfere with other vital cellular pathways? Furthermore, to what extent do knockouts (e.g., *tnaA*, *tyrR*, *pheA*, *tyrA*, *trpB*) impact the system? These issues should be addressed in a more comprehensive and clear manner in the revised manuscript.

In conclusion, the biosynthetic module is well-designed, effectively integrating metabolic engineering, orthogonal translation, and synthetic biology to enable in vivo production of ncAAs. While the presented data demonstrates the feasibility of the approach, further improvements in substrate transport, toxicity assessment, and long-term pathway stability could significantly enhance its potential for industrial applications.

Nonetheless, the manuscript requires major revision due to potential flaws and limitations in the data. First, the uptake and transport of ncAAs are not fully optimized. While intracellular concentrations were measured, the study does not explicitly address the optimization of transporter expression to maximize uptake efficiency. Second, the toxicity of aryl-aldehydes is not addressed in detail. For instance, some aldehydes can be cytotoxic, potentially reducing cell viability at high concentrations. Data correlating cell growth rates with ncAA yields would be invaluable for evaluating toxicity thresholds. Additionally, certain aryl-aldehydes may require specific transporters for efficient import, which has not been thoroughly explored. This reviewer is particularly concerned about the toxicity of aryl-aldehydes, as it is not adequately discussed. For example, providing data on cell growth rates versus ncAA yields would help clarify toxicity thresholds and their impact on the system.

Next, the issue of non-uniform efficiency in ncAA incorporation across proteins must be addressed. While the study demonstrates successful ncAA incorporation, it does not thoroughly discuss the variations in efficiency observed among different model proteins. For instance, certain proteins may present steric hindrance that reduces ncAA incorporation yields, a factor that warrants further exploration.

Finally, there is limited discussion on the long-term stability of the engineered metabolic pathway. Two critical concerns arise: (i) Engineered pathways may become unstable over time due to mutations in long-term cultures, and (ii) The metabolic burden on host cells and plasmid retention rates could impact reproducibility. Including stability data on plasmid retention and the metabolic burden imposed by the pathway would significantly strengthen the study and enhance its applicability for industrial or long-term use. The authors should also consider genomic integration of the engineered enzymes, as orthogonal translation systems have been successfully integrated into the genome in recent studies (e.g. ACS Synth Biol. 13(9), 2992–3002).

Reviewer #2

(Remarks to the Author)

Review Report

1. Significant Findings : This study presents an innovative platform that integrates the biosynthesis of aromatic noncanonical amino acids (ncAAs) with genetic code expansion (GCE) in *E. coli*. The research successfully demonstrates the *in vivo* synthesis and incorporation of 40 different ncAAs, 19 of which were efficiently incorporated into proteins using three orthogonal translation systems. The platform was further validated through the production of macrocyclic peptides and antibodies incorporating functional ncAAs, showcasing its broad applicability in protein engineering and synthetic biology. The ability to directly couple ncAA biosynthesis with GCE within the same host cell represents a significant advancement in the field, addressing key limitations related to the cost and availability of ncAAs in large-scale applications.

2. Significance and Originality : The work is highly relevant to the fields of synthetic biology, protein engineering, and metabolic engineering. By overcoming the dependency on externally supplied ncAAs, this approach significantly enhances the feasibility of large-scale protein production with site-specific modifications. Compared to existing literature, which primarily relies on external supplementation or metabolic engineering with limited substrate scope, this study presents a more efficient and versatile strategy. Previous works, such as those employing *de novo* biosynthesis of ncAAs [Mehl et al., 2003; Chen et al., 2020] or α -keto acid precursors [Jung et al., 2014], have demonstrated partial success, but this study offers a more integrated and scalable solution. The findings align well with recent efforts in genetic code expansion but provide a unique contribution by enabling a broader range of ncAAs to be synthesized and incorporated *in situ*.

3. Support for Conclusions:

The conclusions are well-supported by extensive experimental data, including UPLC-MS quantification, fluorescence measurements, SDS-PAGE, and high-resolution mass spectrometry. The successful incorporation of biosynthesized ncAAs into sfGFP and antibody fragments provides compelling evidence for the platform's functionality.

4. Potential Issues in Data Analysis and Interpretation: While the study provides robust data, a few aspects require clarification:

5. Methodological Soundness and Reproducibility:

The methodology is well-documented, and the use of multiple orthogonal translation systems strengthens the validity of the findings.

Final Recommendation:

This study presents a significant advancement in ncAA biosynthesis and incorporation, with clear potential for applications in protein engineering and biopharmaceutical development. While the overall quality of the research is high, addressing the concerns outlined below will further strengthen the manuscript.

Recommendation: Minor Revision (if above additional metabolic and analyses can be provided).

1. The metabolic burden imposed by the engineered biosynthetic pathways on host cell growth and protein production. Comparative analysis of protein yields with and without *in situ* ncAA biosynthesis to assess whether metabolic trade-offs affect final product titers.
2. Some ncAAs exhibit lower conversion efficiencies, particularly those with ortho-substituents. The manuscript should discuss whether enzyme engineering or pathway optimization could improve their yields.
3. The study mentions that certain aldehyde substrates exhibit cytotoxicity, but a more detailed discussion on how this affects host cell viability and ncAA production would be valuable. Including bacterial growth curves under different substrate conditions could address this concern.
4. Provide detailed enzyme kinetics (e.g., k_{cat}/K_m) for key catalytic steps in ncAA biosynthesis.
5. Include a more comprehensive description of the optimization process for pathway engineering, particularly how enzyme promiscuity and substrate scope were evaluated.
6. In the Introduction section:
 - o Line 45: A missing comma between analytical chemistry and catalysis.
 - o Line 45: A missing reference for medicines.

- o Line 50: A missing reference for β -amino acids.
- o Line 85: Citation format inconsistency—references 33,34 should be carefully checked for alignment with the reference list.
- 7. The manuscript should use antibody fragments rather than antibodies. It is recommended to modify Line 329 to "Biosynthesis of Antibody Fragments with ncAAs in *E. coli*."
- 8. Figure 6C: The reported molecular weight of Anti-HER2-Fab (49,149 Da) does not match the observed molecular weight in the SDS-PAGE gel, showing a significant discrepancy. The same issue applies to J591-Fab. Please provide an explanation.
- 9. For Supplementary Table 1: Expression yield of sfGFP with ncAA modification
- 10. The reported sfGFP-pBoF yield of $64.0 \text{ mg}\cdot\text{L}^{-1}$ should be rechecked for accuracy. The mutation sites of pBoF (Y32S/L65A/H70M/D158S/L162E) were derived from Brustad, E.; Bushey, M.L.; Lee, J.W.; Groff, D.; Liu, W.; Schultz, P.G. *Angew Chem Int Ed Engl* 2008, 47, 8220–8223. Under normal 1 mM ncAA induction, the reported yield appears to be unexpectedly high. Please verify the experimental data.
- 11. The incorporation efficiency of pAzF into proteins is relatively low and should be rechecked. Based on the description and data in the manuscript, when evaluating the compatibility of the ncAA synthesis platform with various OTSs, the biosynthesis of the pAz substrate did not show a significant advantage over the direct supplementation of pAzF, and the protein yield was also low. However, in the production of macrocyclic peptides, pAz exhibited outstanding efficiency, with an intensity 25 times higher than that observed with direct pAzF supplementation. Could the authors briefly explain the potential reasons behind this observation?

Reviewer #3

(Remarks to the Author)

Zhang et al reported the coupling of biosynthesis of non-canonical amino acids with genetic code expansion for producing ncAA-incorporated proteins, antibodies and macrocyclic peptides in *E. coli* cells. While similar systems for generating other types of ncAAs have been developed in recent years, this work has made certain progress from several aspects. First, various types of aromatic ncAAs have been biosynthesized and incorporated into target proteins. Second, the enzymes selected for synthesizing aromatic ncAAs are robust due to their promiscuous catalytic activity. Third, the coupling of biosynthesis and incorporation of ncAAs was demonstrated for producing model proteins (GFP), peptides and antibodies in cells. Nevertheless, the major drawback of this study is that the authors have not demonstrated the significance of their study in solving the major challenges of genetic code expansion technology, i.e., either lowering the cost of commercially available but expensive ncAAs or overcoming the poor uptake obstacle of cell membrane-impermeable ncAAs. They need to precisely select some of the aromatic ncAAs they generated and prove its significance regarding the major challenges. Another concern is that they only demonstrated the successful incorporation of the biosynthesized ncAAs into the target proteins or peptides but did not show the utilization of these ncAAs in protein research and application. For instance, they may present evidence to demonstrate that the incorporated ncAAs with photo-crosslinking activity in the target protein could capture the protein-protein interactions; the fluorescence of ncAAc could be used to probe the conformational change of the target protein in living cells; or some ncAAs could mimic the post-translational modification of the target proteins, and so on.

Minor issue:

It is suggested to include the control protein (the wild type protein without ncAA incorporation) in Mass spectrometry analysis in comparison with the ncAA-incorporated target protein (Figs. 2E, 4E).

Version 1:

Reviewer comments:

Reviewer #1

(Remarks to the Author)

This reviewer finds that the revised manuscript, including both the main text and the supplementary information, presents compelling evidence and newly generated data that strongly support its publication.

Reviewer #2

(Remarks to the Author)

I have carefully evaluated the revised manuscript and the authors' detailed responses to the previous review comments. The authors have thoroughly addressed all concerns raised in the initial round of review, and the revisions have significantly enhanced the clarity, rigor, and impact of the work. The additional data regarding metabolic trade-offs and the mechanistic insights into substrate-specific bottlenecks are well-presented and scientifically sound. The manuscript now provides a complete and compelling demonstration of an integrated platform for *in vivo* biosynthesis and incorporation of aromatic ncAAs, with wide-ranging applications in protein engineering and synthetic biology. I find the current version of the manuscript to be scientifically solid, clearly written, and ready for publication without further revision. Based on the thorough revisions and comprehensive responses provided, I recommend the manuscript for acceptance in its current form.

Reviewer #3

(Remarks to the Author)

In new Figure 6d, the wild type antibody (without A121pAzF) should be included as a control to show the specific

conjugation of the fluorescence probe to pAzF.

Reviewer #1 (Remarks to the Author):

The study by Zhang et al. presents a robust platform for the biosynthesis of aromatic ncAAs and their incorporation into proteins via GCE in *Escherichia coli*, with the goal of enhancing large-scale protein production and expanding protein functionality. The authors successfully demonstrated the biosynthesis and incorporation of 40 distinct aromatic ncAAs, including para-azidophenylalanine (pAzF), para-boronophenylalanine (pBoF), meta-nitrophenylalanine (mNF), para-ethynylphenylalanine (pENF), para-O-propargylphenylalanine (pPrF), and various substituted phenylalanines. This work builds on the well-established concept of coupling orthogonal translation systems (OTSs) with engineered metabolic pathways in *E. coli* to enable efficient in vivo incorporation of ncAAs into target proteins.

The core novelty of this study lies in the application of metabolic engineering to develop a biosynthetic pathway that couples the in-situ production of ncAAs with genetic code expansion. This is achieved through a multi-enzyme cascade that converts aromatic aldehydes into ncAAs, which are then incorporated into modified peptides, enzymes, and antibodies. Specifically, the authors engineered the shikimate pathway - a natural pathway found in microbes, fungi, plants, and protists - to create a multi-enzyme cascade. This pathway typically converts simple carbohydrate precursors, such as phosphoenolpyruvate (PEP) and erythrose-4-phosphate (E4P), into chorismate, a key intermediate for the biosynthesis of aromatic amino acids. The key innovation here is the use of various aromatic substitutions of aryl-aldehydes, which are well tolerated by the system, to generate a library of substituted phenylalanine analogues. These analogues are subsequently incorporated into model proteins, demonstrating the versatility and potential of this approach.

As I understand it, their engineering approach includes three key components: (i) Enhancing endogenous enzymes from the shikimate and phenylalanine biosynthesis pathways; (ii) Introducing heterologous enzymes to extend substrate specificity and improve conversion efficiency; and (iii) Optimizing precursor availability to increase ncAA yields, using *Escherichia coli* as the expression platform. The feasibility of this approach was demonstrated through its application to various model proteins, including Superfolder GFP (sfGFP), macrocyclic peptides, and antibodies (such as the anti-HER2 single-chain variable fragment (scFv), anti-HER2 antigen-binding fragment (Fab), and J591-Fab). Undoubtedly, this strategy significantly simplifies the use of bioorthogonal labeling (bioconjugation) in protein science and protein-based material science, offering a powerful tool for advancing these fields.

Response: We sincerely appreciate the reviewer's thoughtful and insightful comments, as well as their kind consideration of our work for publication. The feedback provided is valuable and constructive, offering crucial guidance that will significantly enhance the quality of our manuscript. We are committed to addressing each point in our revision to ensure the paper meets the highest standards of clarity, rigor, and contribution to the field.

As far as I can assess, the uptake of aryl-aldehydes is not without challenges: the efficiency of uptake is a critical factor determining the overall success of the system. The authors tested forty aldehyde analogues in *E. coli* RARE (DE3) strains harboring the biosynthetic module. However, it remains unclear when the co-expression of transport-enhancing genes is necessary and when passive diffusion alone suffices. The manuscript data suggest that intracellular accumulation of

ncAAs varies depending on the aryl-aldehyde substrate and its metabolic conversion efficiency. Are there any general rules governing this process, or must it be determined on a case-by-case basis?

Response: Thank you very much for your important comment. While we evaluated the potential role of transporters in substrate uptake, no dedicated aldehyde uptake transporters have been characterized to date, to our knowledge. This limitation precluded systematic investigation of transport-enhancing genetic modifications. In parallel, the structural diversity of aryl-aldehyde substrates with varied functional groups required individualized assessment of metabolic conversion efficiency, as substrate-specific interactions with the biosynthetic machinery influenced pathway performance.

Additionally, what intracellular ncAA concentrations (in terms of order of magnitude) are sufficient for efficient incorporation?

Response: Thank you very much for your important comment. The ncAA concentrations for sufficient incorporation varies depending on the efficiency of corresponding Aminoacyl-tRNA synthetases (AARSs). It was reported that 0.5-1 mM ncAA was sufficient for most AARSs. We examined the concentration of pIF as a representative example in this study. The data shown in Supplementary Figure 8 suggested that 1 mM pIF was sufficient for incorporation. This result was discussed in the revised manuscript.

Supplementary Figure 8 GFP production level was related to ncAA concentration. Detection of sfGFP(Y151pIF) fluorescence intensity of strain expressed PpLTA-RpTD, pIFRS/tRNA^{Pyl}_{CUA} pair and sfGFP(Y151TAG). Different *p*-iodobenzaldehyde concentration were added in the culture and induced at 30°C for 24 h. The density of cells and the fluorescence intensity (Ex: 485 nm; Em: 528 nm) of sfGFP with incorporation of biosynthetic pIF was detected by a plate reader.

Could the conversion of aryl-aldehydes to ncAAs produce intermediates that interfere with other vital cellular pathways?

Response: Thank you very much for your important comment. The engineered pathway primarily influences amino acid transamination while exhibiting minimal crosstalk with other cellular metabolic networks. To evaluate the broader physiological effects of pathway intermediates, we purified intermediates from the *p*-iodophenylalanine (pIF) biosynthetic route and assessed their toxicity at 1 mM concentrations. Growth inhibition assays revealed that 1 mM phenylserine reduced

bacterial growth by 10%, whereas *p*-iodophenylpyruvate caused a 20% growth reduction after 12 hours. These findings suggest that while intermediates exhibit moderate cytotoxicity, the pathway's relative metabolic isolation mitigates systemic disruption, supporting its compatibility with host physiology.

Supplementary Figure 7 The impact of intermediates of aryl-aldehydes to ncAAs on cell growth of *E. coli*. *E. coli* RARE (DE3) harboring PpLTA and RpTD. 1 mM *p*-iodobenzaldehyde (**a**), 1 mM *p*-iodophenylserine (**b**), 1 mM *p*-iodophenylpyruvic acid (**c**) or 1 mM *p*-iodophenylalanine (**d**) were added to the culture medium at the same time of IPTG inducing expression when OD₆₀₀ reached 1.0. No reactive components were added to the control group. The strains were then induced at 30 °C for 12 h, and samples were taken every two hours to detect bacterial concentration (OD₆₀₀).

Furthermore, to what extent do knockouts (e.g., *tnaA*, *tyrR*, *pheA*, *tyrA*, *trpB*) impact the system? These issues should be addressed in a more comprehensive and clear manner in the revised manuscript.

Response: Thank you very much for your important comment. The engineered pathway relies on endogenous host aminotransferases to catalyze its final transamination step. To identify the specific enzyme(s) driving non-canonical amino acid (ncAA) production, we systematically evaluated the catalytic activity of three purified *E. coli* aminotransferases, aromatic amino acid aminotransferase (TyrB), aspartate aminotransferase (AspC), and branched-chain amino acid aminotransferase (IlvE), using representative keto-acid intermediates. While TyrB and AspC showed robust catalytic

efficiency, IlvE displayed significantly lower activity (Supplementary Figure 15), suggesting that TyrB and AspC are the primary contributors to the terminal transamination step.

Supplementary figure 15 Synthesis of amino acids from keto acids by purified transaminase. Purified transaminase derived from *E. coli* is used to catalyze the reaction from keto acids to amino acids. 1 mL mixture contained 1 mM phenylpyruvic acid (1% DMSO), 5 mM L-glutamic acid (L-Glu), 0.1 mg·mL⁻¹ ATs in phosphate buffer (50 mM Na₂HPO₄, pH 7.5, 50 mM NaCl). The mixture was stirred at room temperature (25 °C) and detected after 10 min.

In conclusion, the biosynthetic module is well-designed, effectively integrating metabolic engineering, orthogonal translation, and synthetic biology to enable in vivo production of ncAAs. While the presented data demonstrates the feasibility of the approach, further improvements in substrate transport, toxicity assessment, and long-term pathway stability could significantly enhance its potential for industrial applications.

Nonetheless, the manuscript requires major revision due to potential flaws and limitations in the data. First, the uptake and transport of ncAAs are not fully optimized. While intracellular concentrations were measured, the study does not explicitly address the optimization of transporter expression to maximize uptake efficiency.

Response: Thank you very much for your important comment. While we evaluated the potential role of transporters in substrate uptake, no dedicated aldehyde uptake transporters have been characterized to date, to our knowledge. This limitation precluded systematic investigation of transport-enhancing genetic modifications.

Second, the toxicity of aryl-aldehydes is not addressed in detail. For instance, some aldehydes can be cytotoxic, potentially reducing cell viability at high concentrations. Data correlating cell growth rates with ncAA yields would be invaluable for evaluating toxicity thresholds. Additionally, certain aryl-aldehydes may require specific transporters for efficient import, which has not been thoroughly explored. This reviewer is particularly concerned about the toxicity of aryl-aldehydes, as it is not adequately discussed. For example, providing data on cell growth rates versus ncAA yields would help clarify toxicity thresholds and their impact on the system.

Response: Thank you very much for your insightful comment. We systematically analyzed cell growth rates relative to ncAA production yields using four representative aldehyde substrates. As shown in Supplementary Figure 20, toxicity profiles were influenced not only by the aldehydes themselves but also by the interplay between ncAA accumulation rates and their inherent cytotoxicity. For pAzF, optical density (OD₆₀₀ after 12 hours remained comparable to the control

group, indicating minimal growth inhibition. In contrast, other substrates caused significant growth suppression (10–20% reduction) despite moderate to high ncAA yields (50–85%). As proposed in the Discussion, future work could address this toxicity challenge by engineering extended biosynthetic pathways, for example, incorporating enzymes such as transaminases, alcohol dehydrogenases, or carboxylic acid reductases, to maintain low, steady-state aldehyde concentrations via precursor feeding, thereby balancing production efficiency with cellular viability.

Supplementary Figure 20 The impact of aryl-aldehydes on cell growth of *E. coli* and ncAAs yields. *E. coli* RARE (DE3) harboring PpLTA and RpTD. 1 mM *p*-methylbenzaldehyde (pMeBzH) (a), 1 mM *p*-propargyloxybenzaldehyde (pPrBzH) (b), *p*-ethynylbenzaldehyde (pENBzH) (c) or 1 mM *p*-azidebenzaldehyde (pAzBzH) (d), 50 mM Gly and 20 μM PLP were added to the culture medium at the same time of IPTG inducing expression when OD₆₀₀ reached 1.0. No reactive components were added to the control group. The strains were then induced at 30 °C for 12 h, and samples were taken every two hours to detect bacterial concentration (OD₆₀₀) and ncAAs conversion.

Next, the issue of non-uniform efficiency in ncAA incorporation across proteins must be addressed. While the study demonstrates successful ncAA incorporation, it does not thoroughly discuss the variations in efficiency observed among different model proteins. For instance, certain proteins may present steric hindrance that reduces ncAA incorporation yields, a factor that warrants further exploration.

Response: Thank you very much for your insightful comment. Efficient incorporation of non-canonical amino acids (ncAAs) into proteins remains a pervasive challenge in genetic-code expansion research. In this study, we are trying to focus on ncAA biosynthesis, so the protein targets and incorporation sites were all selected based on well-established precedents from the literature.

Nevertheless, the further exploration to large scale production of proteins with ncAAs needs optimization of each steps during the whole process, which includes, but not limited to ncAA biosynthesis, Aminoacyl-tRNA synthetase engineering, codon optimization, host engineering as well as selection of appropriate ncAA incorporation site in the target proteins.

Finally, there is limited discussion on the long-term stability of the engineered metabolic pathway. Two critical concerns arise: (i) Engineered pathways may become unstable over time due to mutations in long-term cultures, and (ii) The metabolic burden on host cells and plasmid retention rates could impact reproducibility. Including stability data on plasmid retention and the metabolic burden imposed by the pathway would significantly strengthen the study and enhance its applicability for industrial or long-term use. The authors should also consider genomic integration of the engineered enzymes, as orthogonal translation systems have been successfully integrated into the genome in recent studies (e.g. ACS Synth Biol. 13(9), 2992–3002).

Response: Thank you very much for your insightful comment. Genomic integration of the engineered enzymes is definitely planned for our future development. The current study aims to establish a generalizable proof-of-concept for in situ synthesis of ncAAs to support genetic code expansion and related research applications. We anticipate that genomic integration will become particularly advantageous when targeting a specific ncAA for dedicated applications, as it enables coordinated optimization and integration of both the biosynthetic pathway and orthogonal translation machinery within a single microbial chassis. This unified approach promises enhanced efficiency, stability, and scalability for tailored ncAA production systems.

Reviewer #2 (Remarks to the Author):

Review Report

1. Significant Findings: This study presents an innovative platform that integrates the biosynthesis of aromatic noncanonical amino acids (ncAAs) with genetic code expansion (GCE) in *E. coli*. The research successfully demonstrates the in vivo synthesis and incorporation of 40 different ncAAs, 19 of which were efficiently incorporated into proteins using three orthogonal translation systems. The platform was further validated through the production of macrocyclic peptides and antibodies incorporating functional ncAAs, showcasing its broad applicability in protein engineering and synthetic biology. The ability to directly couple ncAA biosynthesis with GCE within the same host cell represents a significant advancement in the field, addressing key limitations related to the cost and availability of ncAAs in large-scale applications.

2. Significance and Originality: The work is highly relevant to the fields of synthetic biology, protein engineering, and metabolic engineering. By overcoming the dependency on externally supplied ncAAs, this approach significantly enhances the feasibility of large-scale protein production with site-specific modifications. Compared to existing literature, which primarily relies on external supplementation or metabolic engineering with limited substrate scope, this study presents a more efficient and versatile strategy. Previous works, such as those employing de novo biosynthesis of ncAAs [Mehl et al., 2003; Chen et al., 2020] or α -keto acid precursors [Jung et al., 2014], have demonstrated partial success, but this study offers a more integrated and scalable solution. The

findings align well with recent efforts in genetic code expansion but provide a unique contribution by enabling a broader range of ncAAs to be synthesized and incorporated *in situ*.

3.Support for Conclusions:

The conclusions are well-supported by extensive experimental data, including UPLC-MS quantification, fluorescence measurements, SDS-PAGE, and high-resolution mass spectrometry. The successful incorporation of biosynthesized ncAAs into sfGFP and antibody fragments provides compelling evidence for the platform's functionality.

4.Potential Issues in Data Analysis and Interpretation: While the study provides robust data, a few aspects require clarification:

5.Methodological Soundness and Reproducibility:

The methodology is well-documented, and the use of multiple orthogonal translation systems strengthens the validity of the findings.

Final Recommendation:

This study presents a significant advancement in ncAA biosynthesis and incorporation, with clear potential for applications in protein engineering and biopharmaceutical development. While the overall quality of the research is high, addressing the concerns outlined below will further strengthen the manuscript.

Response: We sincerely appreciate the reviewer's thoughtful and insightful comments, as well as their kind consideration of our work for publication. The feedback provided is valuable and constructive, offering crucial guidance that will significantly enhance the quality of our manuscript. We are committed to addressing each point in our revision to ensure the paper meets the highest standards of clarity, rigor, and contribution to the field.

Recommendation: Minor Revision (if above additional metabolic and analyses can be provided).

1. The metabolic burden imposed by the engineered biosynthetic pathways on host cell growth and protein production. Comparative analysis of protein yields with and without *in situ* ncAA biosynthesis to assess whether metabolic trade-offs affect final product titers.

Response: Thank you very much for your insightful comment. The comparative yields of sfGFP produced with and without *in situ* ncAA biosynthesis are summarized in Fig. 4c and discussed in the corresponding section. While metabolic trade-offs associated with *in situ* biosynthesis can reduce final product titers in certain cases, the approach demonstrated a clear advantage in enhancing sfGFP yields for specific ncAAs, such as *meta*-nitrophenylalanine and *para*-boronophenylalanine. These results highlight the context-dependent utility of *in situ* ncAA biosynthesis, where benefits outweigh metabolic costs for substrates with favorable biosynthetic efficiency and low cytotoxicity.

2. Some ncAAs exhibit lower conversion efficiencies, particularly those with ortho-substituents. The manuscript should discuss whether enzyme engineering or pathway optimization could improve their yields.

Response: Thank you very much for your insightful comment. We have identified the rate limiting step of some ncAAs with low lower conversions (Supplementary figure 16 and 17) and discussed the results in the manuscript:

“We investigated the rate-limiting step for substrates that exhibited low conversion in the culture. UPLC analysis, using substrate and intermediate standards, revealed that the

transaminase was responsible for the low conversions of *ortho*-substituted substrates, leading to the accumulation of α -keto acids (Supplementary Figure 16). Conversely, LTA was the limiting factor for substrates with *para*-substituents, resulting in the residue of starting materials, with no detectable α -keto acids (Supplementary Figure 17). These observations illustrate that the synthetic efficiency of this biosynthetic pathway could be further improved through protein engineering of the key enzymes tailored for specific ncAAs of interest.”

3. The study mentions that certain aldehyde substrates exhibit cytotoxicity, but a more detailed discussion on how this affects host cell viability and ncAA production would be valuable. Including bacterial growth curves under different substrate conditions could address this concern.

Response: Thank you very much for your insightful comment. In response to the same comments from reviewer 1 and 2, we systematically analyzed cell growth rates relative to ncAA production yields using four representative aldehyde substrates. As shown in Supplementary Figure 20, toxicity profiles were influenced not only by the aldehydes themselves but also by the interplay between ncAA accumulation rates and their inherent cytotoxicity. For pAzF, optical density (OD₆₀₀ after 12 hours remained comparable to the control group, indicating minimal growth inhibition. In contrast, other substrates caused significant growth suppression (10–20% reduction) despite moderate to high ncAA yields (50–85%). As proposed in the Discussion, future work could address this toxicity challenge by engineering extended biosynthetic pathways, for example, incorporating enzymes such as transaminases, alcohol dehydrogenases, or carboxylic acid reductases, to maintain low, steady-state aldehyde concentrations via precursor feeding, thereby balancing production efficiency with cellular viability.

Supplementary Figure 20 The impact of aryl-aldehydes on cell growth of *E. coli* and ncAAs yields. *E. coli* RARE (DE3) harboring PpLTA and RpTD. 1 mM *p*-methylbenzaldehyde (pMeBzH)

(a), 1 mM *p*-propargyloxybenzaldehyde (pPrBzH) (b), *p*-ethynylbenzaldehyde (pENBzH) (c) or 1 mM *p*-azidebenzaldehyde (pAzBzH) (d), 50 mM Gly and 20 μ M PLP were added to the culture medium at the same time of IPTG inducing expression when OD₆₀₀ reached 1.0. No reactive components were added to the control group. The strains were then induced at 30 °C for 12 h, and samples were taken every two hours to detect bacterial concentration (OD₆₀₀) and ncAAs conversion.

4. Provide detailed enzyme kinetics (e.g., k_{cat}/K_m) for key catalytic steps in ncAA biosynthesis.

Response: Thank you very much for your insightful comment. We measured the detailed enzyme kinetics of PpLTA, CsLTA and RpLTD with the substrate *p*-(Trifluoromethoxy)benzaldehyde. The results were discussed in the revised manuscript: “We performed detailed kinetic analyses of PpLTA, CsLTA, and RpTD using *para*-(trifluoromethoxy)benzaldehyde as the substrate, enabled by the successful purification of diastereoisomeric phenylserine intermediates. Steady-state kinetic measurements revealed that PpLTA and CsLTA exhibited catalytic efficiencies (k_{cat}/K_m of 34600 M⁻¹s⁻¹ and 18206 M⁻¹s⁻¹ respectively) for the production of (2*S*,3*R*)-*p*-(trifluoromethoxy)phenylserine (Supplementary Figure 13). In contrast, the k_{cat}/K_m value for the generation of the (2*S*,3*S*)-diastereomer were significantly lower: 3621 M⁻¹s⁻¹ (PpLTA) and 13564 M⁻¹s⁻¹ (CsLTA). Subsequent analysis of RpTD demonstrated a similar stereodivergence: the enzyme efficiently processed the (2*S*,3*R*)-phenylserine intermediate to the keto-acid ($k_{cat}/K_m = 65231$ M⁻¹s⁻¹, Supplementary Figure 14) but showed reduced activity toward the (2*S*,3*S*)-isomer ($k_{cat}/K_m = 405$ M⁻¹s⁻¹). Based on these kinetic results, we concluded that the (2*S*,3*S*)-phenylserine intermediate is critical for pathway flux.”

Supplementary figure 13 enzyme kinetics for the aldol reaction in ncAA biosynthesis. Kinetic analysis of PpLTA or CsLTA for *p*-(Trifluoromethoxy)benzaldehyde were monitored by UPLC-MS

with standard curve of the product. For the (2*S*, 3*R*)-product of PpLTA (**a**), the substrate was dissolved in DMSO and diluted to 0.1 mM, 0.2 mM, 0.3 mM, 1 mM, 2 mM, 4 mM. For the (2*S*, 3*S*)-product of PpLTA (**b**), the substrate was dissolved in DMSO and diluted to 0.5 mM, 1 mM, 2 mM, 4 mM, 8 mM, 12 mM. For the (2*S*, 3*R*)-product of CsLTA (**c**), or (2*S*, 3*S*)-product of CsLTA (**d**), the substrate was dissolved in DMSO and diluted to 0.1 mM, 0.2 mM, 0.3 mM, 0.5 mM, 1 mM, 2 mM. The concentration of the substrate glycine was always saturated solution. 10 $\mu\text{g}\cdot\text{mL}^{-1}$ PpLTA or 2 $\mu\text{g}\cdot\text{mL}^{-1}$ CsLTA in phosphate buffer (50 mM Na_2HPO_4 , pH 7.5, 50 mM NaCl). The mixture was stirred at room temperature (25 °C). Sample of the reaction solution were taken every 30 s for analysis. Error bars represent the standard deviation from triple samples.

Supplementary figure 14 enzyme kinetics for the deamination in ncAA biosynthesis. Kinetic analysis of RpTD for *p*-(Trifluoromethoxy)phenylserine were monitored by UPLC-MS with standard curve of the product. Substrate was dissolved in phosphate buffer and diluted to 0.5 mM, 1 mM, 2 mM, 4 mM, 8 mM, 12 mM. For (2*S*, 3*R*)-substrate added 1 $\mu\text{g}\cdot\text{mL}^{-1}$ (**a**) and for (2*S*, 3*S*)-substrate added 5 $\mu\text{g}\cdot\text{mL}^{-1}$ (**b**) RpTD in phosphate buffer (50 mM Na_2HPO_4 , pH 7.5, 50 mM NaCl). The mixture was stirred at room temperature (25 °C). Sample of the reaction solution were taken every 30 s for analysis. Error bars represent the standard deviation from triple samples.

5. Include a more comprehensive description of the optimization process for pathway engineering, particularly how enzyme promiscuity and substrate scope were evaluated.

Response: Thank you very much for your insightful comment. Enzyme promiscuity and substrate scope were systematically evaluated using two complementary approaches. Initial pathway efficiency was assessed through analytical-scale whole-cell catalysis, providing rapid screening of biosynthetic capability. Results from this phase are presented in Supplementary Figure 11 (revised Supporting Information). Following reaction optimization, the pathway was validated under preparative-scale whole-cell catalysis. This approach not only confirmed scalability but also enabled the production of non-commercial ncAAs (Supplementary Figure 10), as detailed in the Substrate Scope of the Biosynthetic Pathway section.

Supplementary Figure 11 Biosynthetic efficiency of nAAs from aromatic aldehyde precursors by lyophilized *E. coli* RARE (CsLTA-RpTD) whole-cell catalyst. In whole-cell catalysis, 1mL reaction mixtures contained 1 mM aromatic aldehydes, 10 mM Gly, 10 mM L-Glu and 10 mg lyophilized cell *E. coli* RARE/pACYC-CsLTA-RpTD in phosphate buffer (50 mM Na₂HPO₄, pH 7.4, 50 mM NaCl) with 10% DMSO. The mixture was stirred at room temperature (25 °C) for 12 h and products were characterized by UPLC-MS. Error bars represented the standard deviation from triple samples.

6. In the Introduction section:

o Line 45: A missing comma between analytical chemistry and catalysis.

Response: Thank you very much for your important comment. It was corrected accordingly.

o Line 45: A missing reference for medicines.

Response: Thank you very much for your important comment. Two references was added for medicine and one was added for material sciences.

o Line 50: A missing reference for β -amino acids.

Response: Thank you very much for your important comment. β -amino acids and α,α -disubstituted amino acids were incorporated in the same work (Ref 16 in the revised manuscript).

o Line 85: Citation format inconsistency—references 33,34 should be carefully checked for alignment with the reference list.

Response: Thank you very much for your important comment. The format of the mentioned references has been modified and the other references were also checked.

7. The manuscript should use antibody fragments rather than antibodies. It is recommended to modify Line 329 to "Biosynthesis of Antibody Fragments with nAAs in *E. coli*."

Response: Thank you very much for your important comments, these sentences and expressions were revised accordingly.

8. Figure 6C: The reported molecular weight of Anti-HER2-Fab (49,149 Da) does not match the observed molecular weight in the SDS-PAGE gel, showing a significant discrepancy. The same issue applies to J591-Fab. Please provide an explanation.

Response: Thank you very much for your important comment. We previously analyzed the samples with SDS-PAGE gel in the regular Tris-Glycine buffer. According to the instruction from our marker supplier (<https://www.beyotime.com/product/P0509M.htm>), The buffer system affects the results and the experiments need to be run in HEPES buffer. We reanalyzed all the proteins during revision and the SDS-PAGE images were revised in all figures. The original images were included in the Source data file. On the other hand, the High-resolution mass data matched the expected mass of modified sfGFP and the antibody fragments.

9. For Supplementary Table 1: Expression yield of sfGFP with ncAA modification

Response: Thank you very much for your important comment. It was corrected accordingly.

10. The reported sfGFP-pBoF yield of 64.0 mg·L⁻¹ should be rechecked for accuracy.

The mutation sites of pBoF (Y32S/L65A/H70M/D158S/L162E) were derived from Brustad, E.; Bushey, M.L.; Lee, J.W.; Groff, D.; Liu, W.; Schultz, P.G. *Angew Chem Int Ed Engl* 2008, 47, 8220–8223. Under normal 1 mM ncAA induction, the reported yield appears to be unexpectedly high. Please verify the experimental data.

Response: Thank you very much for your important comment. We validated the yield of sfGFP-pBoF (64.0 mg mg·L⁻¹). It was within the range (12.8 mg·L⁻¹ -72.2 mg·L⁻¹) of all the ncAA-incorporated sfGFP variants in this study. pBoF was incorporated in Z-domain of staphylococcal protein A in *Angew. Chem. Int. Ed.* **2008**, 47, 8220–8223. The yield was around 15 mg L⁻¹. Therefore, the yield of sfGFP-pBoF is practical because sfGFP is a robustly expressed protein and the *in situ* biosynthesis system was shown to improve the yield of pBoF-incorporated proteins (Fig. 4c)

11. The incorporation efficiency of pAzF into proteins is relatively low and should be rechecked. Based on the description and data in the manuscript, when evaluating the compatibility of the ncAA synthesis platform with various OTSs, the biosynthesis of the pAz substrate did not show a significant advantage over the direct supplementation of pAzF, and the protein yield was also low. However, in the production of macrocyclic peptides, pAz exhibited outstanding efficiency, with an intensity 25 times higher than that observed with direct pAzF supplementation. Could the authors briefly explain the potential reasons behind this observation?

Response: Thank you very much for your insightful comment. We sincerely apologize for the oversight in clarifying the experimental design in our original manuscript. To address this, we provide the following clarification: Prior to demonstrating applications, we systematically investigated the impact of orthogonal translation system (OTS) expression vectors on protein and peptide yields. Our findings revealed that the choice of vector significantly influenced production efficiency. The OTS components (pAzFRS-tRNA) were initially cloned into the pUltra vector for superfolder GFP (sfGFP) production. For macrocyclic peptide synthesis, we redesigned the system by cloning pAzFRS into the pEVOL vector with an additional copy of the pAzFRS gene to enhance expression. During manuscript revision, we confirmed that using the pEVOL vector for pAzFRS expression also improved sfGFP yields compared to the pUltra system.

These results were discussed and highlighted in the revised manuscript, while the vectors were shown in Fig. 5a. "Initial experiments showed low fluorescence intensity in sfGFP when using a single-copy pAzFRS-tRNA system in pUltra vector for pAzF incorporation. To address this, we

employed arabinose-inducible promoters in pEVOL vector to drive the expression of two pAzFRS copies, which significantly improved the production of pAzF-containing proteins (Supplementary Figure 22). The improved performance was further validated in the production of pAzF-containing cyclic peptides, as evidenced by both Fig. 5b and Supplementary Figure 23."

Supplementary Figure 22 Fluorescence intensity of sfGFP containing pAzF with co-expression of two copies of pAzFRS. The fluorescence intensity of sfGFP_{Y151TAG} was measured in *E. coli* RARE (DE3) (CsLTA-RpTD) strains containing pAzFRS with one (a) or two (b) copies of *MjpAzFRS*, after 24 hours of fermentation. The culture was supplemented with either 1 mM nCAAs (PG) or 1 mM aromatic aldehyde substrates (EG). Error bars represented the standard deviation from triple samples.

Supplementary Figure 23 Intracellular synthesis of peptide macrocycles with pUltra vector encoding pAzFRS. a. Plasmid design for coupling pAzF biosynthesis and incorporation into *Npu* of cyclo-CLLFVY. pACYCDuet-1 contained CsLTA and RpTD genes for cascade catalysis, pUltra vector contained mutant of *MjTyrRS*/tRNA pair. tRNA expression was driven by a constitutive promoter, whereas other genes were expressed under IPTG induction. b. Cyclo-CLLFVY, with pAzF at positions 2, 4, or 6, was incorporated by the pAzFRS/tRNA^{Tyr} pair.

Reviewer #3 (Remarks to the Author):

Zhang et al reported the coupling of biosynthesis of non-canonical amino acids with genetic code expansion for producing ncAA-incorporated proteins, antibodies and macrocyclic peptides in *E. coli* cells. While similar systems for generating other types of ncAAs have been developed in recent years, this work has made certain progress from several aspects. First, various types of aromatic ncAAs have been biosynthesized and incorporated into target proteins. Second, the enzymes selected for synthesizing aromatic ncAAs are robust due to their promiscuous catalytic activity. Third, the coupling of biosynthesis and incorporation of ncAAs was demonstrated for producing model proteins (GFP), peptides and antibodies in cells. Nevertheless, the major drawback of this study is that the authors have not demonstrated the significance of their study in solving the major challenges of genetic code expansion technology, i.e., either lowering the cost of commercially available but expensive ncAAs or overcoming the poor uptake obstacle of cell membrane-impermeable ncAAs. They need to precisely select some of the aromatic ncAAs they generated and prove its significance regarding the major challenges.

Response: Thank you very much for your insightful comment. Reducing the production costs of non-canonical amino acids (ncAAs), particularly commercially available yet expensive variants, is a central objective of our research. A key example is *para*-boronophenylalanine (pBoF), which has significant applications in bioconjugation, bioorthogonal protein labeling, and enzymatic catalysis. Our integrated approach, combining ncAA biosynthesis with genetic code expansion GCE, demonstrates clear advantages. For instance, using *para*-borono-benzaldehyde as a precursor, we observed two-fold higher normalized fluorescence in labeled proteins compared to control systems supplemented with pre-synthesized pBoF. Importantly, this strategy addresses cost barriers: *para*-borono-benzaldehyde is priced at 376 RMB per 100 g by a local supplier in Shanghai, making it 100-fold more cost-effective than commercially sourced pBoF (43 RMB per 100 mg from the same supplier). This substantial economic benefit underscores the potential of our biosynthesis-GCE platform to democratize access to high-value ncAAs for diverse biotechnological applications.

Another concern is that they only demonstrated the successful incorporation of the biosynthesized ncAAs into the the target proteins or peptides but did not show the utilization of these ncAAs in protein research and application. For instance, they may present evidence to demonstrate that the incorporated ncAAs with photo-crosslinking activity in the target protein could capture the protein-protein interactions; the fluorescence of ncAAc could be used to probe the conformational change of the target protein in living cells; or some ncAAs could mimic the post-translational modification of the target proteins, and so on.

Response: Thank you very much for your important comment. To demonstrate the application and validate the functionality of the antibody fragments generated in this study, we conjugated anti-HER2-Fab-A121pAzF with AF488-DBCO. The resulting conjugate effectively stained HER2-positive cells, as demonstrated by confocal imaging (Fig. 6d).

Minor issue:

It is suggested to include the control protein (the wild type protein without ncAA incorporation) in Mass spectrometry analysis in comparison with the ncAA-incorporated target protein (Figs. 2E, 4E).

Response: Thank you very much for your important comment. We included the Mass spectrometry analysis results of the wild type proteins in the revised supporting information.

Supplementary Figure 25 High-resolution mass spectrometry of wild type sfGFP and antibody fragments.